# CIRCUITSENSE: A HIERARCHICAL MLLM BENCHMARK BRIDGING VISUAL COMPREHENSION AND SYMBOLIC REASONING IN ENGINEERING DESIGN PROCESS

**Arman Akbari** [*,1], **Jian Gao**[1], **Yifei Zou**[1], **Mei Yang**[1], **Jinru Duan**[1], **Dmitrii Torbunov**[2],
**Yanzhi Wang**[1], **Yihui Ren**[2], **Xuan Zhang**[1]

[1]Northeastern University, [2]Brookhaven National Laboratory

## ABSTRACT

Engineering design operates through hierarchical abstraction from system specifications to component implementations, requiring visual understanding coupled with mathematical reasoning at each level. While Multi-modal Large Language Models (MLLMs) excel at natural image tasks, their ability to extract mathematical models from technical diagrams remains unexplored. We present **CircuitSense**, a comprehensive benchmark evaluating circuit understanding across this hierarchy through 8,006+ problems spanning component-level schematics to system-level block diagrams. Our benchmark uniquely examines the complete engineering workflow: Perception, Analysis, and Design, with a particular emphasis on the critical but underexplored capability of deriving symbolic equations from visual inputs. We introduce a hierarchical synthetic generation pipeline consisting of a grid-based schematic generator and a block diagram generator with auto-derived symbolic equation labels. Comprehensive evaluation of eight state-of-the-art MLLMs, including both closed-source and open-source models, reveals fundamental limitations in visual-to-mathematical reasoning. Closed-source models achieve over 85% accuracy on perception tasks involving component recognition and topology identification, yet their performance on symbolic derivation and analytical reasoning falls below 19%, exposing a critical gap between visual parsing and symbolic reasoning. Models with stronger symbolic reasoning capabilities consistently achieve higher design task accuracy, confirming the fundamental role of mathematical understanding in circuit synthesis and establishing symbolic reasoning as the key metric for engineering competence.

**Project page:** `https://circuitsense-benchmark.github.io`

## 1 INTRODUCTION

Mathematical modeling forms the foundation of all engineering disciplines. Mechanical engineers derive equations of motion to predict system dynamics (Shabana, 2005); optical engineers compute ray transfer matrices to design lens systems (Saleh & Teich, 2007). It is a general practice for electronic engineers to translate circuit schematics into symbolic transfer functions to analytically examine the different aspects of the circuit performance (e.g., noise, stability, sensitivity, etc.). For example, a phase-locked loop (PLL) with insufficient phase margin will oscillate, destroying functionality of the entire integrated electronic system, yet this catastrophic failure can only be predicted through mathematical analysis of the feedback network's poles and zeros (Hanumolu et al., 2004). Across these domains, the ability to translate visual representations such as circuit schematics, optical layouts, or system diagrams into precise mathematical formulations determines engineering success. This visual-to-mathematical reasoning represents a fundamental capability that no current AI system can replicate. However, unlike geometry or physics problems that operate in a single

---

[*]Correspondence to `akbari.ar@northeastern.edu`

representational space, engineering uniquely requires this mathematical translation across multiple levels of hierarchy, from components to subsystems to complete architectures.

While Multi-modal Large Language Models (MLLMs) excel at visual perception tasks, they exhibit a critical limitation: the inability to derive symbolic equation from visual representations (Liu et al., 2025; Lu et al., 2021; Pan et al., 2025). This failure is not merely technical but fundamental: equation derivation distinguishes true engineering comprehension from pattern matching. Existing visual circuit benchmarks (Skelic et al., 2025; Shi et al., 2025; Li et al., 2025; Zhao et al., 2025), focus primarily on recognition-based tasks like identifying component types, answering basic multiple-choice questions, or performing shallow numerical calculations. The core capability that defines circuit understanding remains untested: the ability to extract mathematical relationships from visual circuit topology that is consistent across multiple system hierarchies.

We focus on analog circuits as a particularly rich domain for evaluating visual-to-mathematical capabilities. The analog design process progresses through multiple stages: topology creation (determining device types and interconnections) (Lai et al., 2025; Chang et al., 2024; Dong et al., 2023), device sizing (optimizing physical dimensions for performance) (Wang et al., 2020; Lyu et al., 2017; Cao et al., 2024), and layout design (Xu et al., 2019; Kunal et al., 2019; Crossley et al., 2013) (representing circuit as geometric shapes and physical layers for fabrication), with each stage building upon the previous. The design process in analog circuits suffers from long cycles where catastrophic failures like instability, oscillation, and excessive noise often remain hidden until final verification stages. The key to accelerating analog design lies in early mathematical analysis.

This preventive approach relies on translating visual schematics into mathematical models. For example, for low frequency circuit design such as operational amplifier (Op-Amp), engineers derive equations to predict frequency response (Kamath et al., 1974), input and output referred noise (Hillbrand & Russer, 2003), ensure stability margins, and optimize performance trade-off. For radio frequency circuits such as low noise amplifiers and power amplifiers, people derive the circuit's input and output impedance to ensure impedance matching and optimal power transfer (Wang et al., 2010; Nguyen et al., 2004). Yet no existing benchmark evaluates whether AI systems possess this circuit understanding and symbolic reasoning capability. Without examining the symbolic derivation process, we cannot assess whether models truly understand circuits or merely memorize visual patterns, and consequently, whether they can genuinely assist human designers in accelerating design cycles and catching critical failures before costly fabrication. This gap prevents us from determining if MLLMs are ready to serve as engineering tools or remain sophisticated but superficial pattern matchers.

Moreover, circuit design inherently operates across multiple levels of abstraction, requiring engineers to seamlessly navigate between system architecture and component implementation. Engineers typically begin with high-level block diagrams to architect complex systems such as analog-to-digital converters (ADCs), phase-locked loops (PLLs), or multi-stage operational amplifiers. They then systematically decompose these architectural blocks into component-level schematics, implementing each functional block using transistors, op-amps, and passive elements. Figure 1 illustrates this hierarchical decomposition through a PLL example, showing how system-level blocks translate into transistor-level implementations. Appendix A.3 provides detailed analysis questions for both this PLL and a two-stage op-amp, demonstrating the multi-level reasoning required for comprehensive circuit understanding. Despite the critical importance of this hierarchical reasoning capability, no existing benchmark evaluates the ability to bridge between block diagrams and circuit schematics.

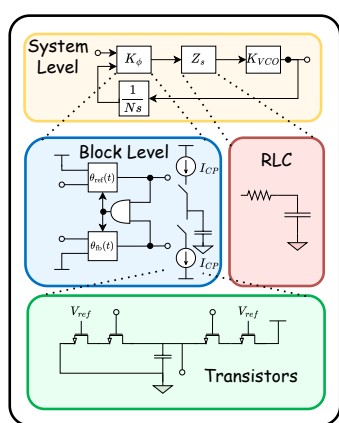

Figure 1: Multi-level hierarchy of Phase Lock Loop Design.

To fill this gap, we propose **CircuitSense**, the first benchmark that systematically evaluates circuit understanding through hierarchical mathematical reasoning. CircuitSense comprises 8,006 problems organized across six hierarchy levels from resistor networks to system-level block diagrams with open-ended and multiple-choice formats, testing three task categories that mirror the engineering workflow: Perception, Analysis, and Design. Our benchmark combines 2,986 carefully

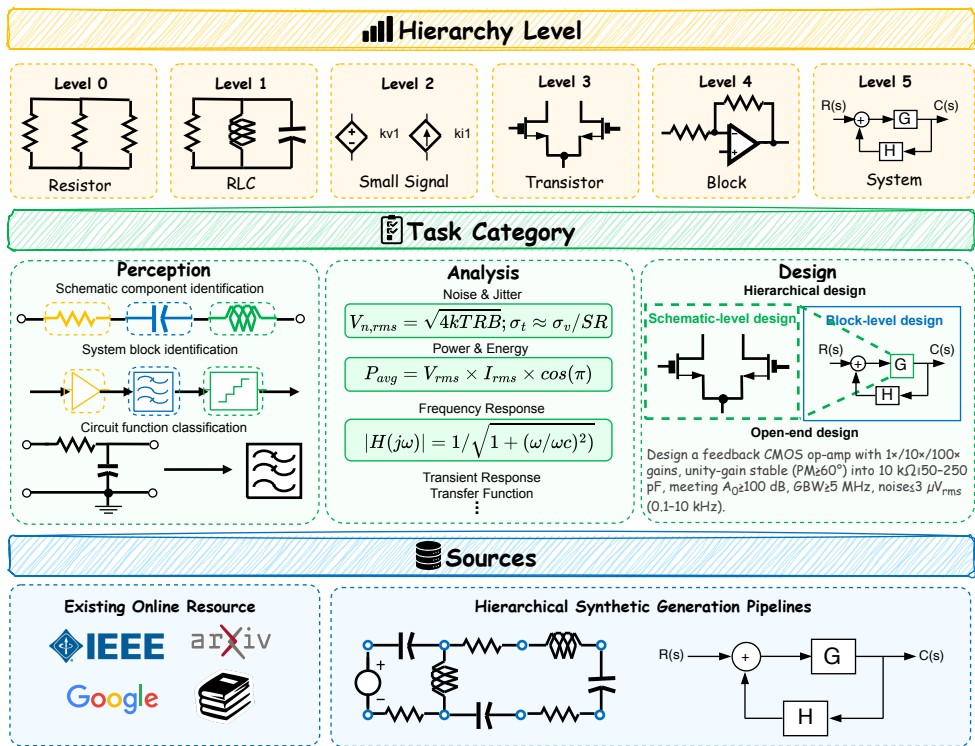

Figure 2: Benchmark overview. CircuitSense evaluates circuit systems understanding across six hierarchy levels (resistor networks to system block diagrams), three task category (Perception, Analysis with equation derivation, and Design), using both curated problems and synthetically generated circuits systems with ground-truth symbolic equations.

curated problems from authoritative textbooks and documents with 5,020 synthetically generated circuits, uniquely emphasizing symbolic derivation. We introduce a hierarchical synthetic generation pipeline consisting of a circuit schematic generator with guaranteed symbolic ground-truth equations, and a block diagram generator with symbolic transfer function ground-truth. This dual approach ensures both component-level depth and system-level breadth while preventing dataset contamination.

As illustrated in Figure 3, we evaluated CircuitSense over 8 state-of-the-art MLLMs and Gemini-2.5-Pro Google DeepMind (2025) showed the best performance among all tasks. Our main contribution and findings can be summarized as:

- **First Multi-Level Visual-to-Analytical Benchmark:** We introduce the first benchmark that systematically evaluates understanding across engineering abstraction levels, from system-level block diagrams to component-level schematics, testing how models connect visual patterns at different scales to their mathematical representations.

- **Hierarchical Synthetic Generation Pipeline:** We developed two synthetic generation pipeline producing samples with guaranteed ground-truth equations: (i) component-level circuits with controlled complexity progression, and (ii) system-level block diagrams with hierarchical feedback structures, enabling isolated evaluation of visual comprehension and mathematical reasoning at each abstraction level.

- **Extensive Multi-Scale Performance Analysis:** Through systematic evaluation of eight state-of-the-art MLLMs and detailed analysis of derivation attempts, we demonstrate that while closed-source models achieve over 85% accuracy on perception tasks, they fail catastrophically at symbolic analysis (below 19% accuracy), with specific bottlenecks identi-

fied including systematic output impedance misinterpretation and algebraic manipulation errors. Our experiments confirm that models with stronger equation derivation capabilities consistently achieve higher design task performance, establishing mathematical understanding as prerequisite for AI-assisted circuit synthesis.

## 2 CIRCUITSENSE

In this section we introduce **CircuitSense**, a comprehensive visual benchmark consisting of 8,006+ problems for evaluating visual circuit understanding across different task categories and abstraction levels. CircuitSense evaluates visual circuit understanding through a hierarchical framework that mirrors the complete engineering design process, from high-level system architecture to detailed component implementation. As shown in Figure 2, the benchmark is organized along two primary axes: task categories and hierarchy levels. The dataset spans three task categories: Perception (890), Analysis (7043), and Design (157), with Analysis comprising the majority of problems as it directly tests the critical capability of extracting mathematical models from visual circuits. Problems are distributed across six hierarchical levels from basic resistor networks to system-level block diagrams, enabling fine-grained assessment of where visual-to-mathematical translation fails.

### 2.1 DATA COLLECTION

We gather 2,986 curated problems for from authoritative sources to ensure broad topical coverage across CMOS analog, RLC network analysis, and system-level circuit design. For Analysis task, our collection drew from two primary categories: (1) canonical textbooks widely adopted in undergraduate and graduate curricula including Gray (2009); Razavi (2017); Allen & Holberg (2012); Bruun; Rahmani-Andebili (2022); Salam & Rahman (2018); (2) university course repositories, including University of Toronto ECE331 (*Analog Electronics*), Georgia Tech ECE6412 (*Analog Integrated Circuit Design*), and Georgia Tech ECE3050 (*Analog Electronics*). For Perception questions we used a subset of circuit images from AnalogGenie (Gao et al., 2025) and our hierarchical synthetic generation pipeline. For Design task, we collected data from canonical analog circuit design textbooks such as Gray (2009); Razavi (2017); Allen & Holberg (2012); Bruun, along with representative problem sets curated from university courses and design problems from ZeroSim (Yang et al., 2025). More detailes are provided in Appendix A.4.

However, curated problems suffer from potential dataset contamination and rarely test equation derivation systematically. To ensure unbiased evaluation, we developed a hierarchical synthetic generation pipeline producing novel circuits with guaranteed ground-truth equations across different hierarchy levels, detailed in Section 2.2.

### 2.2 HIERARCHICAL SYNTHETIC GENERATION PIPELINE

**Circuit Schematic Generator** Our schematic generator extends the MAPS framework (Zhu et al., 2025a) for Linear Pure Resistive Circuits (LPRC) to support the full spectrum of analog components. We construct circuit on an $m \times n$ grid where dimensions are sampled from a Discrete Probability Distribution to ensure topological diversity. Component selection follows a hierarchical probability distribution that differs between inner and outer edges. We support 18 component types organized by complexity: passive elements (R, L, C), sources (voltage, current), controlled sources (VCVS, VCCS, CCVS, CCCS), and active devices (ideal op-amps). We treat the ideal op-amps as template subcircuits (input resistor, feedback network, and high-gain VCVS) and randomly place the whole template in the grid.

The generator enforces electrical validity through multiple constraints: eliminating floating nodes, ensuring at least degree-2 connectivity for all nodes, and maintaining exactly one voltage source per circuit to guarantee a well-defined reference. The grid topology is translated into SPICE-compatible netlists through systematic node labeling and component enumeration. Circuit validation occurs at three levels: topological verification ensures no shorted components and proper control relationships, SPICE simulation confirms DC operating points and AC responses, and symbolic analysis through Lcapy (Hayes, 2022) extracts ground-truth transfer functions $H(s) = V_{out}(s)/V_{in}(s)$ and nodal equations via Modified Nodal Analysis. To manage computational complexity, we imple-

Table 1: Benchmark statistics.

| Task | Subcategory | Count |
|---|---|---|
| *Perception* | | *806* |
| | Component Detection | 200 |
| | Connection Identification | 200 |
| | Function classification | 406 |
| *Analysis* | | *7043* |
| | Frequency Response | 184 |
| | Transient Response | 3811 |
| | Transfer Function Analysis | 1736 |
| | Small Signal Analysis: | 915 |
| | CMR & PSRR | 54 |
| | Noise & Jitter Analysis | 121 |
| | Power & Energy Analysis | 222 |
| *Design* | | *157* |
| | Schematic-level | 63 |
| | Block-level | 56 |
| | Hierarchical | 38 |
| **Total** | | **8,006** |

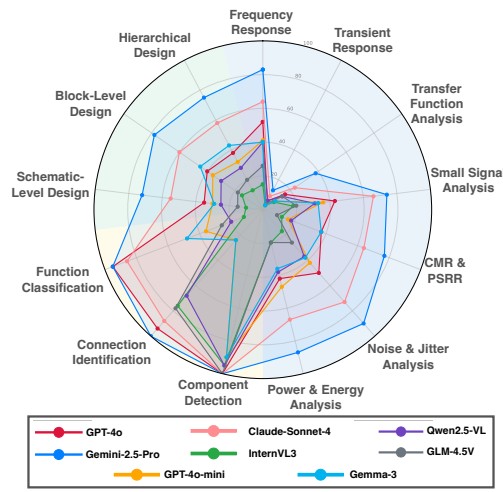

Figure 3: Result of 8 representative MLLMs on Perception, Analysis, and Design tasks.

ment adaptive timeouts based on circuit complexity scores, bypassing symbolic analysis for circuits exceeding practical computation limits.

**Block Diagram Generator** The block diagram pipeline constructs control systems through a two-phase approach that ensures both structural validity and mathematical consistency. We begin by constructing a main signal path consisting of $n \in [\tau_b, \tau_e]$ components, selected from a library of standard transfer functions, placed sequentially along a fixed horizontal axis. Components include transfer function blocks and summing junctions (with randomly assigned sign conventions for each input port), selected with probability [a:b], enabling both positive/negative feedback and feedforward configurations. System complexity increases through systematic addiction of feedback and feedforward paths, with $n_{fb} \in [0, \tau_{fb}]$ feedback loops and $n_{ff} \in [0, \tau_{ff}]$ feedforward paths, with the algorithm preventing duplicate connections through set-based tracking. Each auxiliary path has probability $p_{block} = 0.5$ of containing an intermediate block, generating diverse architectures from simple unity feedback to complex multi-loop systems found in industrial applications such as ADCs and PLLs.

We compute the overall system transfer function using Mason's gain formula (Mason, 1953), which systematically handles multiple feedback loops and forward paths. The algorithm identifies: all forward path $P_k$ from input to output, all loops $L_i$ in the system, and non-touching loop combinations. The system determinant $\Delta = 1 - \Sigma L_i + \Sigma L_i.L_j - \Sigma L_i.L_j.L_k + \dots$ is computed symbolically, wehere the sum includes all combinations of non-touching loops. The overall transfer function becomes $H(s) = \frac{\Sigma P_k.\Delta_k}{\Delta}$, where $\Delta_k$ is the determinant excluding loops that touch forward path $P_k$. This approach correctly handles complex topologies incluing nested loops.

This hierarchical approach to synthetic generation ensures comprehensive coverage from low-level component interactions to high-level system behavior, providing the multi-scale evaluation necessary for assessing true circuit understanding.

## 2.3 BENCHMARK STATISTICS

CircuitSense comprises 8,006 problems organized across three primary task categories that mirror the engineering workflow: Perception (890 problems), Analysis (7,043 problems), and Design (157 problems). As shown in Table 1, the Analysis task dominates our benchmark, reflecting the central importance of mathematical reasoning in circuit understanding. Within Analysis, we balance 2,023 curated problems with 5,020 synthetically generated circuits to ensure both educational validity and protection against dataset contamination. The distribution across subcategories reflects our emphasis on equation derivation capabilities: Transient Response (3,811 problems) and Transfer Function Analysis (1,736 problems) are significantly larger categories because our synthetic gen-

eration pipeline primarily produces circuits for testing these fundamental mathematical skills. The remaining subcategories provide comprehensive coverage of circuit analysis: Small Signal Analysis (915) tests linearization and AC modeling, while Power & Energy Analysis (222), Frequency Response (184), Noise & Jitter Analysis (121), and CMR & PSRR (54) evaluate specialized analytical skills.

Perception tasks comprise three subcategories: Component Detection (200 problems) tests whether models can accurately count and identify circuit elements, Connection Identification (200 problems) evaluates netlist conversion capabilities to verify structural understanding of circuit topology, and Function Classification (406 problems) assesses whether models can infer circuit purpose from visual inspection alone. Together, these tasks establish whether models possess the visual comprehension necessary for subsequent mathematical analysis. Design tasks progress through increasing levels of hierarchy: from schematic-level (63) and block-level design (56) to hierarchical design (38) that requires coordinating multiple abstraction levels.

Problems are organized across six complexity levels that capture the natural progression of circuit design: Level 0 (Resistive Networks, 1,777 samples) for DC analysis, Level 1 (RLC Circuits, 3,147 samples) for frequency-domain reasoning, Level 2 (Small Signal, 537 samples) with controlled sources, Level 3 (Transistor, 795 samples) for device-level analysis, Level 4 (Block, 559 samples) for operational amplifier abstraction, and Level 5 (System Diagrams, 228 samples) for system-level transfer functions. This hierarchical structure enables precise identification of where visual-to-mathematical translation fails as complexity increases. Detailed level-specific statistics and problem distributions are provided in Appendix Table 9.

## 3 EXPERIMENTS

In this section, we conduct comprehensive experiments to evaluate different closed-source and open-source MLLMs on CircuitSense across task categories and hierarchy levels. We test eight state-of-the-art models: Gemini-2.0-Pro (Google DeepMind, 2025), Claude-4-Sonnet (Anthropic, 2025), GPT-4o, GPT-4o-mini, InternVL-3-78B (Zhu et al., 2025b), Qwen2.5-VL-72B-Instruct (Bai et al., 2025), GLM-4.5V (Team et al., 2025b), and Gemma-3-27B (Team et al., 2025a).

### 3.1 EVALUATION FRAMEWORK

We employ two evaluation strategies depending on problem format. For multiple-choice questions, we use exact answer matching after standardized formatting. To enable multiple-choice evaluation on originally open-ended problems, we use Gemini-2.5-Flash to generate three plausible distractor choices plus "None of the above" to avoid forcing random selection. For open-ended questions requiring numerical or short-form answers, we employ LLM-as-a-judge evaluation where Gemini-2.5-Flash compares model responses against ground truth, accounting for equivalent representations and unit conversions, determining correctness based on mathematical equivalence rather than exact string matching. For design tasks that require simulations, we simulate them by Ngspice (ngspice Development Team, 2024) with Skywater 130nm PDK (SkyWater Technology Foundry, 2020).

Evaluating symbolic mathematical expressions presents unique challenges since a single equation can be represented in numerous algebraically equivalent forms. For instance, $H(s) = 1/(RCs + 1)$ is mathematically identical to $H(s) = (1/RC)/(s + 1/RC)$. To address this, we implement a rigorous symbolic comparison pipeline using SymPy (Meurer et al., 2017) that performs: (1) parsing both predicted and ground-truth equations into symbolic expression trees, (2) algebraic simplification, (3) verification through symbolic subtraction, and (4) numerical validation by evaluating both expressions at 100 random complex frequency points when symbolic comparison is computationally intractable. This multi-pronged approach ensures robust evaluation even when models produce correct but differently formatted equations.

### 3.2 MAIN RESULTS

**Perception Task** We evaluated models on three perception subtasks: Component Detection, Connection Identification, and Function Classification. As Table 2 shows, closed-source models demonstrate strong visual understanding with over 86% accuracy, confirming that perception is not the

Table 2: Perception task results

| Model | Component Detec.(%) | Connection Ident.(%) | Function Class.(%) |
|---|---|---|---|
| GPT-4o | 100 | 70 | 95 |
| GPT-4o-mini | 90 | 24 | 36 |
| Gemini-2.5-Pro | **100** | **100** | **95** |
| Claude-Sonnet-4 | 100 | 88 | 86 |
| InternVL3-72B | 95 | 76 | 12 |
| Qwen2.5-VL | 95 | 68 | 20 |
| GLM-4.5V | 100 | 78 | 26 |
| Gemma-3-27B | 90 | 24 | 68 |

Table 3: Design task results

| Model | Schematic-level(%) | Block-level(%) | Hierarchical-Design(%) |
|---|---|---|---|
| GPT-4o | 10.52 | 36.36 | 18.92 |
| GPT-4o-mini | 30.64 | 36.36 | 32.43 |
| Gemini-2.5-Pro | **36.38** | **67.27** | **51.35** |
| Claude-Sonnet-4 | 17.54 | 51.83 | 29.83 |
| InternVL3-72B | 7.01 | 52.73 | 29.73 |
| Qwen2.5-VL | 8.76 | 30.91 | 18.92 |
| GLM-4.5V | 15.79 | 50.91 | 32.35 |
| Gemma-3-27B | 29.03 | 45.45 | 43.24 |

Table 4: Accuracies of different models on Analysis subcategories.

| Model | Frequency Response | Transient Response | Transfer Function Analysis | Small Signal Analysis | CMR & PSRR | Noise & Jitter Analysis | Power & Energy Analysis |
|---|---|---|---|---|---|---|---|
| GPT-4o | 52 | 6 | 16 | 43 | 37 | 50 | 42 |
| GPT-4o-mini | 41 | 4 | 9 | 36 | 16 | 42 | 47 |
| Gemini-2.5-Pro | **83** | **13** | **38** | **74** | **77** | **90** | **87** |
| Claude-Sonnet-4 | 64 | 9 | 23 | 66 | 64 | 73 | 67 |
| InternVL3-78B | 15 | 3 | 8 | 18 | 12 | 17 | 20 |
| Qwen2.5-VL-72B-Instruct | 40 | 6 | 14 | 31 | 18 | 37 | 38 |
| GLM-4.5V | 26 | 4 | 14 | 20 | 9 | 26 | 20 |
| Gemma-3-27B | 40 | 3 | 9 | 33 | 37 | 38 | 36 |

bottleneck for these systems. GPT-4o and Gemini-2.5-Pro achieve near-perfect performance (94-100%), while Claude-Sonnet-4 maintains solid accuracy above 85%. In contrast, open-source models struggle significantly with basic circuit structure recognition. For instance, GLM-4.5V achieves only 26% on Function Classification and 78% on Connection Identification, suggesting fundamental limitations in visual processing capabilities that precede any mathematical reasoning challenges.

**Analysis Task** We examined performance of models across Analysis subcategories. Table 4 reveals that Gemini-2.5-Pro dominates across all categories (13-90%), followed by GPT-4o and Claude-Sonnet-4 (6-73%), while open-source models struggle significantly (below 40%). Furthermore, models achieve higher accuracy on traditionally complex tasks like Noise & Jitter Analysis (up to 90%) and Power & Energy Analysis (up to 87%) compared to fundamental tasks like Transient Response (3-13%) and Transfer Function Analysis (8-38%). This counterintuitive result occurs because our synthetic problems are concentrated in these two fundamental subcategories, exposing the critical gap between memorized textbook solutions and genuine mathematical understanding. When models cannot rely on pattern matching from training data and must derive equations from novel circuits, their performance collapses dramatically. Section 4 separates synthetic and curated results to demonstrate how this weakness manifests when models cannot rely on memorized patterns.

**Design Task** Table 3 reveals a clear hierarchical pattern in design capabilities across all models. Models demonstrate significantly stronger performance at block-level design (30.91-67.27%) compared to schematic-level design (7.01-36.38%), with hierarchical design falling between these extremes. This pattern indicates that models can more readily manipulate abstract functional blocks than translate specifications into detailed component-level implementations. Notably, Gemini-2.5-Pro, which demonstrated superior symbolic equation derivation capabilities in the Analysis tasks, also dominates the Design tasks with 36.38% schematic-level, 67.27% block-level, and 51.35% hierarchical design accuracy. This correlation between symbolic reasoning and design performance suggests that equation derivation capability serves as a fundamental prerequisite for circuit synthesis.

Table 5: Results of curated problems for Analysis task with multiple choice and open-ended format.

| Model | Level 0 (Resistor) | Level 1 (RLC) | Level 2 (Small Signal) | Level 3 (Transistor) | Level 4 (Block) | Overall Accuracy |
|---|---|---|---|---|---|---|
| *Multiple Choice Format (%)* | | | | | | |
| GPT-4o | 39.80 | 49.58 | 32.88 | 48.80 | 39.58 | 45.07 |
| GPT-4o-mini | 43.88 | 49.58 | 26.03 | 23.20 | 45.83 | 36.49 |
| Claude-Sonnet-4 | 66.72 | 71.22 | 61.64 | 72.01 | 66.67 | 69.67 |
| Gemini-2.5-Pro | **74.04** | **87.39** | **78.08** | **81.72** | **89.58** | **80.71** |
| InternVL3-78B | 23.16 | 20.59 | 13.70 | 13.11 | 14.58 | 18.06 |
| Qwen2.5-VL-72b-instruct | 29.53 | 41.60 | 30.14 | 35.94 | 29.17 | 34.90 |
| GLM-4.5V | 24.63 | 29.20 | 9.59 | 17.28 | 31.25 | 22.42 |
| Gemma-3-27B | 33.28 | 35.08 | 9.59 | 38.21 | 25.00 | 34.59 |
| *Open-ended format (%)* | | | | | | |
| GPT-4o | 29.59 | 29.83 | 19.18 | 13.96 | 17.81 | 22.84 |
| GPT-4o-mini | 44.44 | 38.24 | 32.88 | 18.62 | 31.25 | 32.09 |
| Claude-Sonnet-4 | 35.56 | 50.21 | 12.33 | 27.04 | 33.33 | 34.76 |
| Gemini-2.5-Pro | **76.98** | **84.87** | **73.97** | **55.85** | **72.92** | **70.32** |
| InternVL3-78B | 20.79 | 19.54 | 6.85 | 14.47 | 10.42 | 17.26 |
| Qwen2.5-VL-72B-Instruct | 28.73 | 31.30 | 16.44 | 13.71 | 22.92 | 22.85 |
| GLM-4.5V | 34.44 | 39.71 | 13.70 | 19.50 | 25.00 | 28.83 |
| Gemma-3-27B | 24.60 | 25.84 | 8.22 | 10.69 | 8.33 | 18.44 |

Table 6: Performance comparison on our hierarchical synthetic problems with symbolic equation ground truth.

| Model | Level 0 (Resistor) | Level 1 (RLC) | Level 2 (Small Signal) | Level 4 (Block) | Level 5 (System) | Overall |
|---|---|---|---|---|---|---|
| GPT-4o | 1.50 | 3.33 | 5.80 | 7.33 | 9.65 | 4.98 |
| GPT-4o-mini | 3.20 | 3.80 | 7.54 | 3.40 | 0.88 | 3.76 |
| Claude-Sonnet-4 | 2.83 | 5.16 | 5.80 | 11.64 | 7.89 | 6.29 |
| Gemini-2.5-Pro | **3.49** | **11.67** | **38.00** | **12.33** | **35.96** | **19.06** |
| InternVL3-78B | 1.50 | 3.67 | 6.68 | 3.72 | 0.44 | 3.50 |
| Qwen2.5-VL-72B-Instruct | 0.83 | 4.17 | 6.03 | 6.64 | 10.09 | 4.96 |
| GLM-4.5V | 0.33 | 7.33 | 4.00 | 4.50 | 5.70 | 4.09 |
| Gemma-3-27B | 0.80 | 3.00 | 6.03 | 4.60 | 4.82 | 3.84 |

## 4 DISCUSSION

### 4.1 CURATED VS. SYNTHETIC PERFORMANCE

While overall Analysis task performance provided initial insights, the aggregate 7,043-problem evaluation masks critical patterns that emerge only through systematic decomposition. We conducted three complementary evaluations six-level hierarchy. First, we tested 2,023 curated textbook problems in multiple-choice format, where models could leverage answer elimination strategies. Second, we evaluated the same problems in open-ended format, removing the scaffolding of provided options. Third, we assessed 5,020 synthetic circuits requiring direct equation derivation without any answer choices.

The three evaluation formats reveal systematic degradation in mathematical reasoning capability. As shown in Table 5, in multiple-choice format, Gemini-2.5-Pro achieves 80.71% on curated problems and maintains 70.32% in open-ended evaluation. This 10-point drop suggests some analytical ability beyond elimination. However, other models collapse catastrophically without answer options, falling below 35% and exposing heavy reliance on pattern matching. Table 6 shows that on synthetic circuits requiring equation derivation, it catastrophically fails at 19.06%, a 61-percentage-point drop from multiple-choice performance. Other models show even steeper degradation: Claude-Sonnet-4 falls from 69.67% (multiple-choice) to 34.76% (open-ended) to just 6.29% (synthetic), while open-

Table 7: Performance Analysis on Gemini-2.5-Pro across transfer function derivation subtasks.

| Subtask | Description | Acc. (%) |
|---|---|---|
| Component Identification | Identify all components | 97 |
| Impedance Assignment | Convert components to s-domain impedance | 95 |
| Total Impedance Calculation | Compute equivalent input impedance | 81 |
| Output Impedance Derivation | Calculate the impedance at the output node | 8 |
| Impedance Ratio Formation | Apply voltage divider principle correctly | 39 |
| Transfer Function Simplification | Simplify to canonical form | 55 |

source models barely exceed 4% on synthetic problems. This systematic collapse confirms that models rely on answer elimination and pattern matching rather than mathematical reasoning.

Analysis across hierarchy levels reveals that models develop specialized capabilities rather than uniform understanding. On curated problems (Table 5), different models excel at different abstraction levels. Gemini-2.5-Pro peaks at Level 4 (Block-level with ideal op-amps, 89.58%), while Claude-Sonnet-4 achieves highest performance at Level 3 (Transistor circuits, 72.01%). This specialization pattern persists in synthetic evaluation (Table 6) but with revealing differences: Gemini-2.5-Pro achieves its best synthetic performance at Level 2 (Small Signal, 38.00%) and Level 5 (System-level block diagrams, 35.96%), while Claude-Sonnet-4 peaks at Level 4 (Block-level, 11.64%).

## 4.2 FAILURE POINT ANALYSIS

To understand why models fail at equation derivation despite recognizing circuit components, we analyzed 100 transfer function derivation attempts by Gemini-2.5-Pro, decomposing the process into six sequential subtasks. As shown in Table 7, while 55% of attempts correctly computed total impedance, only 8% succeeded at output impedance derivation which is seemingly a simpler task. This 47% drop represents the primary bottleneck in the entire pipeline. The subsequent partial recovery to 39% in impedance ratio formation and 55% in final transfer function suggests the model sometimes reaches correct answers through different reasoning paths compare to human. The higher final accuracy compared to Table 4 reflects our selection of the first 100 questions, which proved easier than the full transfer function analysis subset. Additional system-level failure analysis is provided in Appendix A.1

## 5 RELATED WORKS

**Visual Reasoning in Multi-modal Language Models**   Recent advances in Multi-modal Large Language Models (MLLMs) (Bai et al., 2025; Zhu et al., 2025b; Google DeepMind, 2025) have demonstrated remarkable progress in integrating visual and linguistic information, achieving strong performance on tasks like visual question answering. To evaluate these capabilities, several benchmarks have emerged focusing on visual mathematical reasoning. Most visual math benchmarks (Lu et al., 2024; Wang et al., 2025) evaluates mathematical reasoning in visual contexts but primarily tests knowledge-centric problems that can often be solved through pattern recognition rather than true mathematical understanding. Scientific diagram benchmarks including ScienceQA (Lu et al., 2022), and SeePhy (Xiang et al., 2025) extend evaluation to domain-specific content, testing understanding of physics phenomena. However, these benchmarks evaluate whether models can select correct answers or perform numerical calculations, but do not assess the fundamental capability of translating visual representations into formal symbolic mathematical expressions.

**Visual Circuit Understanding Benchmarks**   Existing circuit-focused benchmarks severely underestimate the complexity of circuit analysis by focusing on shallow tasks within single abstraction levels. CIRCUIT (Skelic et al., 2025) present 510 RLC-only circuit questions. EEE-Bench Li et al. (2025) propose 2860 hand-picked visual questions across electrical and electronics engineering. MMCircuitEval Zhao et al. (2025) also curated 3614 question-answer pairs for digital and analog circuits. Furthermore, AMSbench (Shi et al., 2025) provides analog and mixed-signal circuit problems but focuses on multiple-choice questions testing conceptual understanding. Critically, none of these benchmarks evaluate the fundamental capability of translating visual circuit representations

into symbolic mathematical equations and also lack hierarchical organization that mirrors real engineering abstraction levels. CircuitSense addresses this gap by systematically evaluating visual comprehension and mathematical reasoning across the complete hierarchy.

# 6 CONCLUSION

We introduce CircuitSense, a comprehensive benchmark of 8,006 problems for evaluating visual-to-mathematical reasoning in circuit understanding which combines curated questions with synthetic problems focused on symbolic equation derivation. Our hierarchical synthetic generation pipeline produces novel circuits across six levels with guaranteed ground-truth symbolic equations, enabling rigorous evaluation. Our extensive evaluation on perception, analysis, and design tasks shows that models demonstrate adequate perception (85%+ for closed-source) but fail catastrophically at mathematical symbolic modeling (below 19%). This mathematical weakness directly undermines their design capabilities.

# 7 LIMITATION AND FUTURE WORKS

While CircuitSense advances circuit understanding evaluation, several limitations present opportunities for expansion. Our synthetic pipeline currently focuses on transfer function derivation and nodal analysis, missing other analysis types like noise analysis or frequency response. We plan to extend our symbolic generation pipeline to all subcategories. Computational constraints limit synthetic circuits to 12-15 components since symbolic equation derivation becomes prohibitively expensive beyond this scale, restricting our ability to test understanding of larger circuits.

# 8 REPRODUCIBILITY STATEMENT

To ensure the reproducibility of our results, we provide comprehensive resources and documentation. The complete CircuitSense dataset is included in the supplementary materials. Our hierarchical synthetic generation pipeline code for creating additional circuits with guaranteed ground-truth equations is available at GitHub. The exact model versions and inference parameters for all six evaluated MLLMs are specified in Appendix A.5. Furthermore, the llm-as-a-judge prompt template is also provided in Appendix A.8 fro evaluation process.

# 9 ACKNOWLEDGMENTS

We thank anonymous reviewers for their valuable feedback. This project is partially supported by the National Science Foundation under awards CCF-2526432 and CCF-2416375. Yihui Ren and Dmitrii Torbunov were supported by the Laboratory Directed Research and Development Program of Brookhaven National Laboratory, which is operated and managed for the U.S. Department of Energy Office of Science by Brookhaven Science Associates under contract No. DE-SC0012704.

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

Table 8: System level analysis

| Models | Exact | High-level |
|--------|-------|------------|
| GPT-4o | 9.65 | 1.75 |
| GPT-4o-mini | 0.88 | 10.09 |
| Claude-Sonnet-4 | 7.89 | 28.51 |
| Gemini-2.5-Pro | **35.96** | **39.04** |
| InternVL3-78B | 0.44 | 1.75 |
| Qwen2.5-VL-72B | 10.09 | 9.65 |
| GLM-4.5V | 5.70 | 17.98 |
| Gemma-3-27B | 4.82 | 12.72 |

Table 9: Analysis: Detailed Statistics by Level

| Abstraction Levels | Curated Data | Synthetic Data |
|--------------------|--------------|----------------|
| Level 0 (Resistor) | 631 | 1,146 |
| Level 1 (RLC) | 476 | 2,671 |
| Level 2 (Small Signal) | 73 | 464 |
| Level 3 (Transistor) | 795 | - |
| Level 4 (Block) | 48 | 511 |
| Level 5 (System) | - | 228 |
| **Total** | **2,023** | **5,020** |

# A  APPENDIX

## A.1  BLOCK DIAGRAM TRANSFER FUNCTION ANALYSIS

**Symbolic vs. Abstract Representations** To understand how models handle system-level abstraction, we analyzed their performance on block diagram transfer function derivation under three conditions: high-level representations using simplified block labels ("$G_1$" or "$H_1$"), exact representations with complete transfer functions "$10/(s + 5)$". This comparison reveals whether models struggle with topological understanding of feedback systems or with the algebraic manipulation required for symbolic computation.

Table 8 demonstrates that algebraic complexity, not topological reasoning, fundamentally limits model performance. All models show significant accuracy degradation when moving from high-level to exact representations: Gemini-2.0-Pro drops from 38.18% to 35.96%, while Claude-Sonnet-3.5 exhibits a more dramatic decline from 28.51% to 7.89%. This consistent pattern reveals that models can successfully apply Mason's gain formula to abstract symbols but fail when manipulating complex rational functions with multiple terms. The performance gap indicates that current MLLMs possess adequate understanding of feedback topology and control theory principles but lack the symbolic mathematics capabilities essential for engineering analysis, suggesting that improvements should focus on enhancing algebraic reasoning rather than visual comprehension.

## A.2  HIERARCHY STATISTICS

We classified each question according to its component complexity, from Level 0 (resistor-only networks) to Level 5 (system-level block diagrams). Table 9 presents the distribution across abstraction levels for both curated and synthetic datasets within the Analysis task category. The curated data concentrates at the extremes—Level 0 (631) and Level 3 (795)—reflecting textbook emphasis on foundational concepts and transistor-level design. Notably, we have no curated Level 5 problems and no synthetic Level 3 problems, as system-level textbook problems rarely require equation derivation while transistor circuits resist symbolic generation due to their nonlinear device models.

## A.3  DETAILED HIERARCHICAL EXAMPLES

We present two concrete examples of design questions that require answers derived from multiple levels of analysis. These examples show the application of symbolic equation derivation across different hierarchy in engineering design process which motivated us to collect CircuitSense.

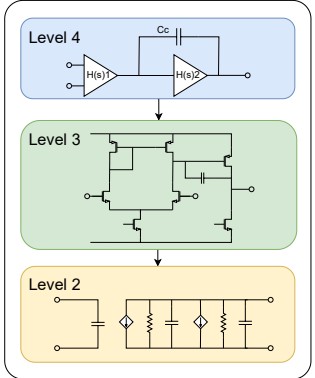

Figure 4: Step-by-step analysis of Two-stage Op-Amp

### A.3.1 TWO-STAGE OP-AMP

Figure 4 illustrates how Levels 4 through 2 can be applied to analyze a two-stage Op-Amp's Gain transfer function, with each level yielding concrete analytical outputs . At the block level (Level 4), the amplifier is identified as two cascaded gain stages with a compensation capacitor, leading to the high-level transfer expression as shown below:

$$\frac{V_{\text{out}}(s)}{V_{\text{in}}(s)} \approx H(s)_1 \cdot H(s)_2 \cdot N_{Cc}(s) \tag{1}$$

Where $H(s)_{1,2}$ is the transfer function of corresponding stage. $N_{Cc}(s)$ is the Zero factor introduced by feedback capacitor. Next, at the transistor level (Level 3), the actual circuit topology is considered: the differential input pair with current-mirror load feeding into a common-source second stage. At this level, At this level, key device parameters of stage 1 and stage 2, such as the transconductance ($g_m$) and output resistance ($r_o$), can be derived:

$$g_{m1,2} = \frac{\partial I_D}{\partial V_{GS}} \tag{2}$$

$$r_{o1,2} \approx \frac{1}{\lambda I_D} \tag{3}$$

Where, $I_D$ is the drain current; $V_{GS}$ is the gate-to-source voltage; $\lambda$ is the channel-length modulation parameter. Once the key parameters and transistor structure are clear, it can be abstracted into a small-signal equivalent (Level 2), each MOSFET is replaced by its hybrid-$\pi$ model, reducing the circuit to dependent sources, $r_o$, and parasitic capacitance. After involving the pole and zero effects, the final transfer function can then be written explicitly as:

$$\frac{V_{\text{out}}(s)}{V_{\text{in}}(s)} \approx H(s)_1 \cdot H(s)_2 \cdot N_{Cc}(s) \approx \frac{g_{m1}r_{o1}}{\left(1 + \frac{s}{\omega_{p1}}\right)} \cdot \frac{g_{m2}r_{o2}}{\left(1 - \frac{s}{\omega_z}\right)} \cdot \left(1 + \frac{s}{\omega_{p2}}\right) \tag{4}$$

This flow connects qualitative structure identification to quantitative expressions, ensuring that the final analysis yields a concrete transfer function.

### A.3.2 PHASE-LOCKED LOOP

Another example of hierarchical analysis is loop gain transfer function of a PLL system. As mentioned in the introduction section, Figure 1 illustrates the hierarch of the PLL, where we focus on the Frequency & Phase Detector (PFD) and the Low-Pass Filter (LPF). At Level 5 (system level), the PLL is partitioned into its main functional blocks: PFD, LPF, Voltage-Controlled Oscillator (VCO), and Frequency Divider, and the loop gain in the phase domain can be expressed as:

$$L(s) = K_\phi \cdot Z(s) \cdot K_{vco} \cdot \frac{1}{s\,N} \tag{5}$$

Moving to Level 4, the PFD is recognized as consisting of two flip-flops, one AND gate, and a charge pump. It acts like a phase-to-current gain:

$$K_\phi = \frac{I_{cp}}{2\pi} \cdot \Delta\phi(t) = \frac{I_{cp}}{2\pi} \cdot (\theta_{\text{ref}}(t) - \theta_{\text{fb}}(t)) \tag{6}$$

Where, $\Delta\phi(t) = \theta_{\theta_{\text{ref}}(t) - \theta_{\text{fb}}(t)}$ is the phase error, and $\theta_{\text{ref.fb}}$ represents reference and feedback phase; To obtain the exact charge pump current $I_{cp}$, we zoom into the level-3 transistor schematic of the charge pump within the PFD. At this level, $I_{cp}$ can be explicitly derived from the transistor equations as:

$$I_{\text{cp}} \approx \tfrac{1}{2}\, \mu_n C_{\text{ox}}\, \frac{W_{\text{NMOS}}}{L_{\text{NMOS}}}\, V_{ov,\text{NMOS}}^2 \tag{7}$$

The same hierarchical analysis framework can be applied to the VCO and the Frequency Divider transfer function analysis. The LPF is modeled as a simple level 1 RLC network,

$$Z(s) = \frac{1}{sC} \tag{8}$$

This hierarchical flow results a complete and quantitative framework for analyzing PLL close loop gain.

## A.4 Data Collection Details

Aside from the sources we mentioned in Section 2 we also have collected data from communities and online platforms dedicated to circuit design (Learn Electronics India; AIC Design; All About Circuits; Embedded Wala; Chegg Inc.). Each problem underwent verification by graduate students with circuit design knowledge.

To standardize the representation of problems across these diverse sources, we developed an offline Flask-based tool, Circuit Benchmark Sample Creator (CBSC), that provides a structured interface for manual data entry and organization. Using CBSC, we separately inserted problem components including circuit diagrams, difficulty levels, source information, questions, answers, and step-by-step derivations. Once the content was entered and submitted, the tool automatically generated a well-structured folder system to store and index the problems. This workflow ensured that all benchmark entries maintained consistent formatting and organization while preserving the integrity of the original materials.

We structure the benchmark by considering a folder for each question consisting of `q#_question.txt` which have the text part of the question, `q#_image.png` which is the image, `q#_ta.txt` which save the ground-truth, `q#_mc.txt` which holds the multiple choice if the question requres, `q#_a.txt` which contains the correct choice. The question folder also includes `q#_der.txt` which is the step-by-step solution and `q#_category.txt` which is the subcategories for Analysis task.

## A.5 Experiment Models Details

All experiments were conducted using the following model versions and parameters:

- **GPT-4o**: `gpt-4o-2024-08-06` (snapshot date: August 6, 2024)
- **GPT-4o-mini**: `openai/gpt-4o-mini`
- **Gemini-2.5-Pro**: `gemini-2.5-pro-preview-0605` (preview version: June 5, 2025)
- **Claude-Sonnet-4**: `claude-4-sonnet`
- **InternVL3-78B**: Official release version 3.0
- **Qwen2.5-VL-72B-Instruct**: Instruction-tuned version 2.5
- **GLM-4.5V**: Vision-enabled version 4.5
- **Gemma-3-27B**: `google/gemma-3-27b-it`

Inference Parameters:

- Temperature: 0.1 (for all models to ensure consistency and reproducibility)
- Maximum tokens: 4096
- Top-p: 0.95 (where applicable)

## A.6 SYNTHETIC EXAMPLES

### Synthetic Example Q1: Nodal Equation

**Question:** Derive the nodal equation for node 2 in the s-domain. Express the equation using only the circuit elements and their values as labeled in the diagram. Make sure the final answer is just the symbolic equation Vn2(s) = ..., where the right side contains only the labeled components and sources from the circuit diagram.

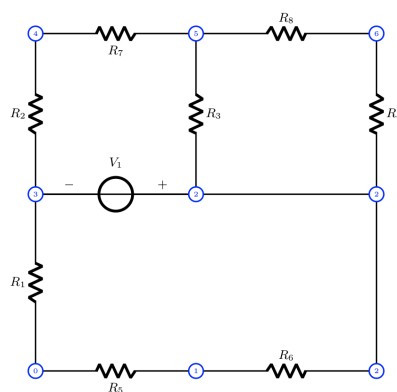

**Corresponding Netlist:**

```
R5 1 0 R5
R1 0 3 R1
R6 1 2 R6
V1 2 3 V1
R2 3 4 R2
R3 5 2 R3
R4 6 2 R4
R7 5 4 R7
R8 5 6 R8
```

**Ground-truth:**
$$Vn2(s) = V1 * (R5 + R6)/(s * (R1 + R5 + R6))$$

Claude-Sonnet-4:
$$Vn2(s) = V3 + V1$$

Gemini-2.5-Pro:
$$Vn2(s) = V1 * (R5 + R6)/(R1 + R5 + R6)$$

GPT-4o:
$$V_2(s) = \frac{\frac{V_5(s)}{R3} + \frac{V_6(s)}{R4} + \frac{V_1(s)}{R6} - \frac{V1}{R2}}{\frac{1}{R3} + \frac{1}{R4} + \frac{1}{R6}}$$

InternVL3-78B:
$$Vn2(s) = (V5(s)/R3 + V6(s)/R4 + V1(s)/R6 + V3(s)/R8)/(1/R3 + 1/R4 + 1/R6 + 1/R8)$$

Qwen2.5-VL-72B-Instruct:
$$Vn2(s) = \frac{\frac{Vn1(s)}{R6} + \frac{Vn3(s)}{R2}}{\frac{1}{R3} + \frac{1}{R4} + \frac{1}{R6} + \frac{1}{R2}}$$

GLM-4.5V:
$$V_{n2}(s) = \frac{\frac{V_{n1}(s)}{R_6} + \frac{V_{n4}(s)}{R_2} + \frac{V_{n5}(s)}{R_3} + \frac{V_{n6}(s)}{R_4} + V_1(s)\left(\frac{1}{R_1} + \frac{1}{R_2}\right)}{\frac{1}{R_1} + \frac{1}{R_2} + \frac{1}{R_3} + \frac{1}{R_4} + \frac{1}{R_6}}$$

## Synthetic Example Q2: RLC Transfer Function

**Question:** Derive the nodal equation for node 3 in the s-domain. Express the equation using only the circuit elements and their values as labeled in the diagram. Make sure the final answer is just the symbolic equation Vn3(s) = ..., where the right side contains only the labeled components and sources from the circuit diagram.

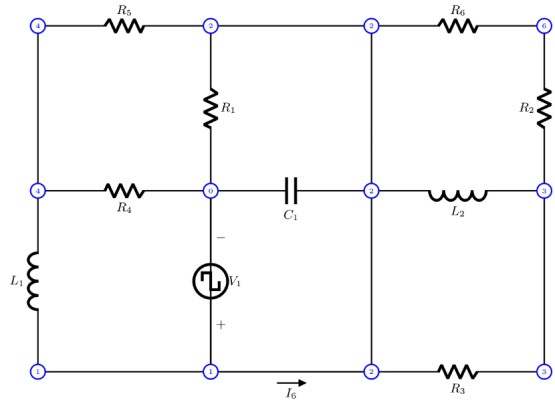

**Netlist Extraction:**

```
L1 1 4 L1
V1 1 0 step
R3 3 2 R3
R4 0 4 R4
C1 2 0 C1
R1 2 0 R1
L2 3 2 L2
R2 3 6 R2
R5 4 2 R5
R6 2 6 R6
```

**Ground-turth:** Vn3(s) = R1*R4*V1/(s*(C1*(R1*(L1*R4*s**2 + L1*R5*s**2 + R4*R5*s)) + L1* (R1*s + R4*s + R5*s) + R4*(R1 + R5))

Claude-Sonnet-4:

$$V3(s) = (V6(s) * s * L2 * R3 + V2(s) * R2 * (R3 + s * L2))/(s * L2 * (R2 + R3) + R2 * R3)$$

Gemini-2.5-Pro:

$$Vn3(s) = V1(s)$$

GPT-4o:

$$V_3(s) = \frac{\frac{V_6(s)}{R_2} + \left(\frac{1}{sL_2} + \frac{1}{R_3}\right) V_2(s)}{\frac{1}{R_2} + \frac{1}{sL_2} + \frac{1}{R_3}}$$

InternVL3-78B:

$$Vn3(s) = (V6(s) * (1/R2) + V2(s) * (1/R3 + 1/(s * L2)))/(1/R2 + 1/R3 + 1/(s * L2))$$

Qwen2.5-VL-72B-Instruct:

$$Vn3(s) = (Vn2(s)/R2 + Vn1(s)/R3)/(1/R2 + 1/R3 + 1/(s * L2))$$

GLM-4.5V:

$$V_{n3}(s) = \frac{sL_2R_3V_{n6} + R_2(sL_2 + R_3)V_{n2}}{sL_2(R_2 + R_3) + R_2R_3}$$

### Synthetic Example Q3: Small Signal (Dependent Sources)

**Question:** What is the transfer function from V1 to R1 in this circuit?

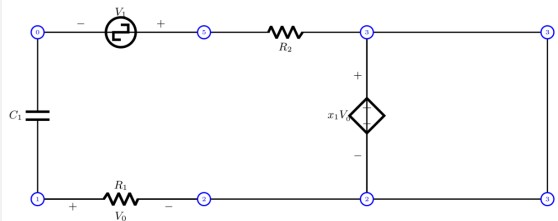

**Netlist Extraction:**

```
R1 1 2 R1
C1 1 0 C1
E1 3 2 1 2 x_1 0
V1 5 0 step
R2 3 5 R2
```

**Ground-truth:**

$$H(s) = ((R1 * s/(R1 * x_1 - R1 - R2))/(s - 1/(C1 * R1 * x_1 - C1 * R1 - C1 * R2))) * 1$$

Claude-Sonnet-4:

$$H(s) = (R1 * s * C1)/((1 + x1) * R1 * s * C1 + 1)$$

Gemini-2.5-Pro:

$$H(s) = (-s * R1 * C1)/(1 + s * C1 * (R1 * (1 - x1) + R2))$$

GPT-4o:

$$H(s) = \frac{sC_1}{sC_1 + \frac{x_1}{R_1}}$$

InternVL3-78B:

$$H(s) = 1/(1 + s * C1 * R1 * (1 - x1))$$

Qwen2.5-VL-78B-Instruct:

$$H(s) = \frac{\frac{1}{R_2} + V_2\left(\frac{1}{R_1} - \frac{1}{R_2}\right)}{\frac{1}{R_1} - s * C_1 - \frac{x_1}{R_2}}$$

GLM-4.5V:

$$H(s) = \frac{sC_1 R_1}{sC_1(R_2 + R_1(x_1 - 1)) + 1}$$

---

**Synthetic Example Q5: Ideal Op-amp**

**Question:** Derive the nodal equation for node 3 in the s-domain. Express the equation using only the circuit elements and their values as labeled in the diagram. Make sure the final answer is just the symbolic equation Vn3(s) = ..., where the right side contains only the labeled components and sources from the circuit diagram.

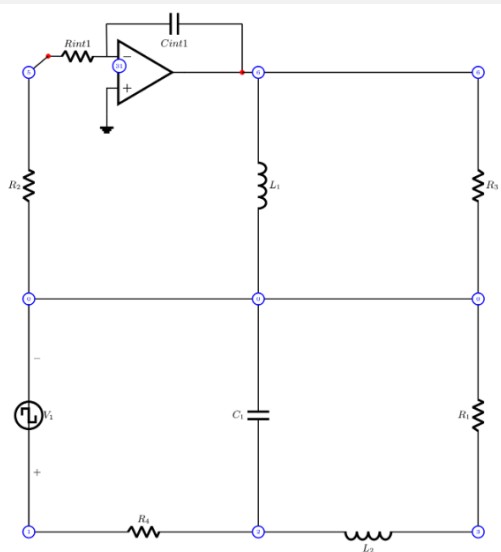

**Netlist Extraction:**

```
R4 1 2 R4
V1 1 0 step
L2 3 2 L2
C1 2 0 C1
R1 3 0 R1
R2 0 5 R2
L1 6 0 L1
R3 6 0 R3
Rint1 5 31 Rint1
Cint1 6 31 Cint1
Eint1 6 0 0 31 Ad 0
```

**Ground-truth:**

$$V n3(s) = R1 * V1/(s * (C1 * L2 * R4 * s * *2 + C1 * R1 * R4 * s + L2 * s + R1 + R4))$$

Gemini-2.5-Pro:

$$V n3(s) = (R1 * V n2(s) + s * L2 * V n7(s))/(R1 + s * L2)$$

Claude-Sonnet-4:

$$V n3(s) = (V2(s) * R1 * s * L2 + V5(s) * R4 * R1)/(R1 * R4 + s * L2 * (R1 + R4))$$

GPT-4o:

$$V_3(s) = \left( \frac{V_0(s)}{R1} + \frac{V_2(s)}{sL2} + \frac{V_2(s)}{R4} \right) \left( \frac{1}{R1} + \frac{1}{sL2} + \frac{1}{R4} \right)^{-1}$$

Qwen2.5-VL-72B:

$$V n3(s) = V n2(s)/(1 + R1/(s * L2) + R1/R3)$$

GLM4.5V:

$$V_{n3}(s) = \frac{R_1 V_1}{(R_1 + R_4) + s(L_2 + C_1 R_4 R_1) + s^2(C_1 R_4 L_2)}$$

## Synthetic Example Q6: Enhanced Control System

**Question:** What is the Transfer function of the provided block diagram?

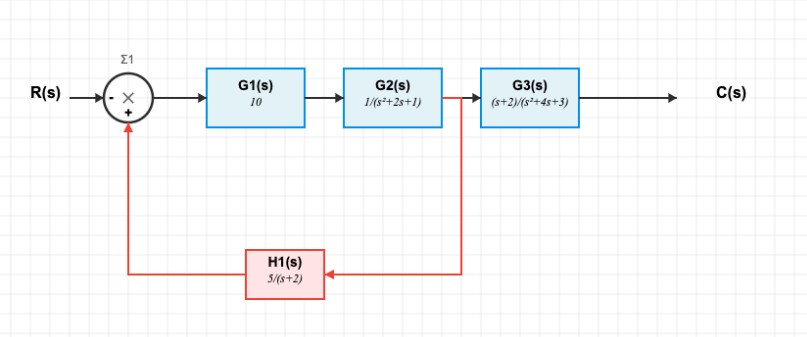

**Ground-truth:**

$$\frac{C(s)}{R(s)} = \frac{(\frac{-10}{(s^2+2s+1)})}{(1 - \frac{50}{(s^2+2s+1).(s+2)})}$$

Claude-Sonnet-4:

$$C(s)/R(s) = 10(s+2)/[(s+1)^3(s+2)^2(s+3)+50]$$

Gemini-2.5-Pro:

$$C(s)/R(s) = (10s^2 + 40s + 40)/(s^5 + 8s^4 + 24s^3 + 84s^2 + 223s + 156)$$

GPT-4o:

$$C(s)/R(s) = \frac{10(s+2)}{s^6 + 2s^5 + s^4 + 2s^3 + s^2 + 6s + 54}$$

InternVL3-78B:

$$C(s)/R(s) = \frac{20(s+2)}{(s^2 + 2s + 1)(s+3)(s+4) + 100s(s+2)/(s+6)}$$

Qwen2.5-VL-72B:

$$\frac{C(s)}{R(s)} = \frac{10(s+2)}{(s^2 + 2s + 1)(s+1)(s+3) + 50}$$

GLM-4.5V:

$$C(s)/R(s) = \frac{10(s+2)}{(s+1)^3(s+2)(s+3) + 50}$$

## Design Example Q6: Simulation-needed Schematic-level question

**Question:** Design the sizing and biasing voltage of an Op-Amp in SKY130nm (VDD 1.8 V) as shown in the provided circuit image.

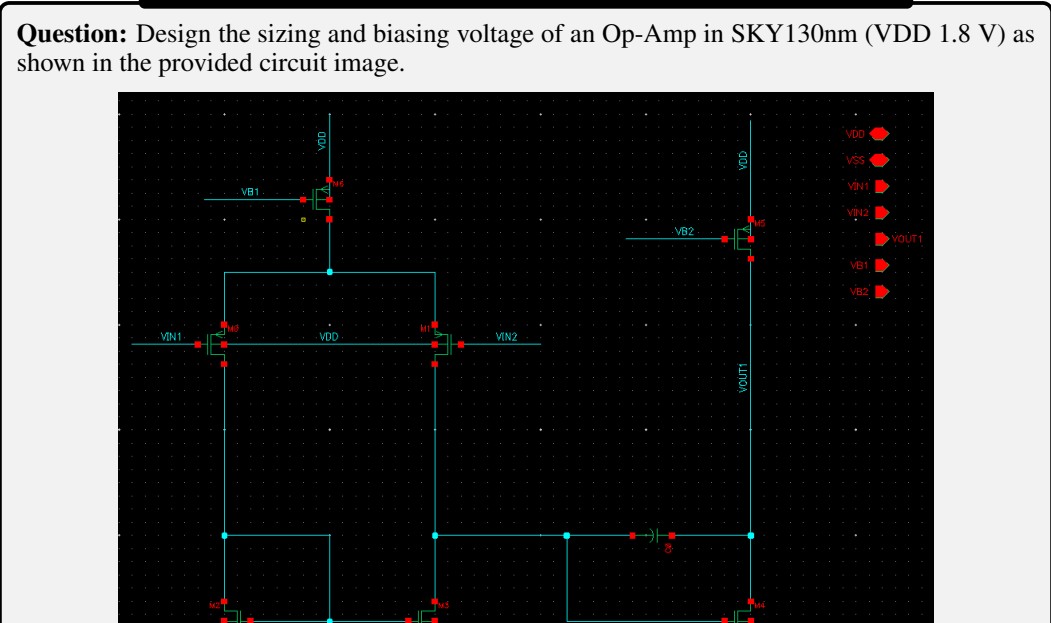

A.7   CURATED PROBLEMS

**Question:**

This problem deals with the op amp shown in the provided circuit image. All device lengths are $1\,\mu m$, the slew rate is $\pm 10\,V/\mu s$, the $GB$ is $10MHz$, the maximum output voltage is $+2V$, the minimum output is $-2V$, and the input common mode range is from $-1V$ to $+2V$. Design all $W$ value of all transistors in this op amp. Your design must meet or exceed the specifications. Ignore bulk effects in this problem.

**Image:**

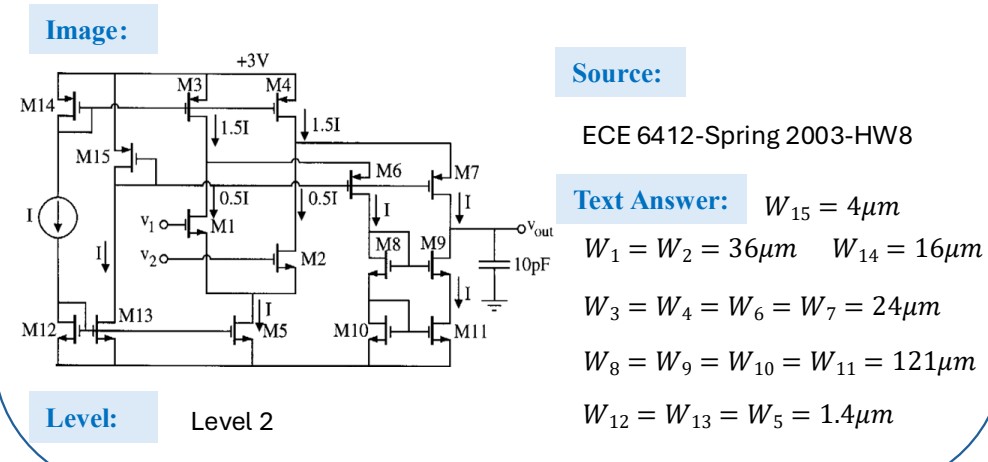

**Source:**

ECE 6412-Spring 2003-HW8

**Text Answer:**   $W_{15} = 4\mu m$

$W_1 = W_2 = 36\mu m$      $W_{14} = 16\mu m$

$W_3 = W_4 = W_6 = W_7 = 24\mu m$

$W_8 = W_9 = W_{10} = W_{11} = 121\mu m$

**Level:**   Level 2

$W_{12} = W_{13} = W_5 = 1.4\mu m$

Example 1: schematic-level design

**Question:**

Design a DPLL using the tri-state topology seen in the provided circuit image that generates a clock signal at a frequency of $100MHz$ from a $50\ MHz$ square wave input. This application of the DPLL is called frequency synthesis.

**Image:**

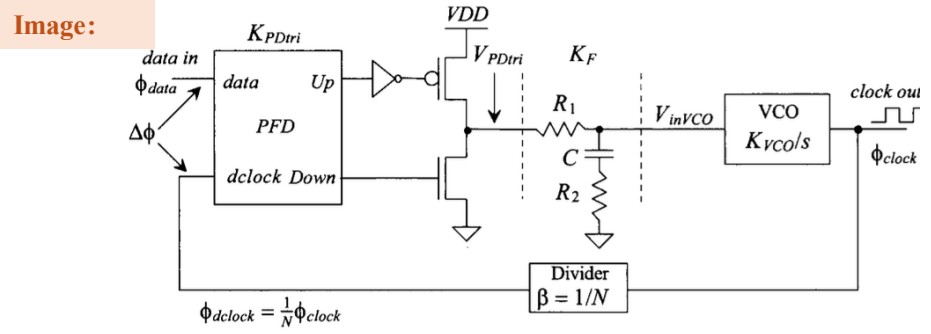

**Source:** CMOS Circuit Design, Layout, and Simulation
P577
Example 19.4

**Level:** Level 2

**Text Answer:** $C = 10pF, R_2 = 20k\Omega, R_1 = 42.5k\Omega$

Example 2: block-level design

**Question:**

Design a CMOS operational amplifier powered from a single $5V$ supply in which all MOSFET channel lengths are fixed at $L = 1\mu m$ and every device operates in saturation; choose the width $W$ of every transistor so that the amplifier meets or exceeds the following specifications: slew rate = $\pm 10V/\mu s$ maximum and minimum output voltage $V_{out(max)} = 4V$, $V_{out(min)} = 1V$, input common-mode range $V_{IC(min)} = 1.5V$ to $V_{IC(max)} = 4V$. And unity-gain bandwidth(GB) =10 MHz; ignore bulk/body effects. Provide a summary table (round each to the nearest micron) listing the $W$ of every transistor.

**Image:**

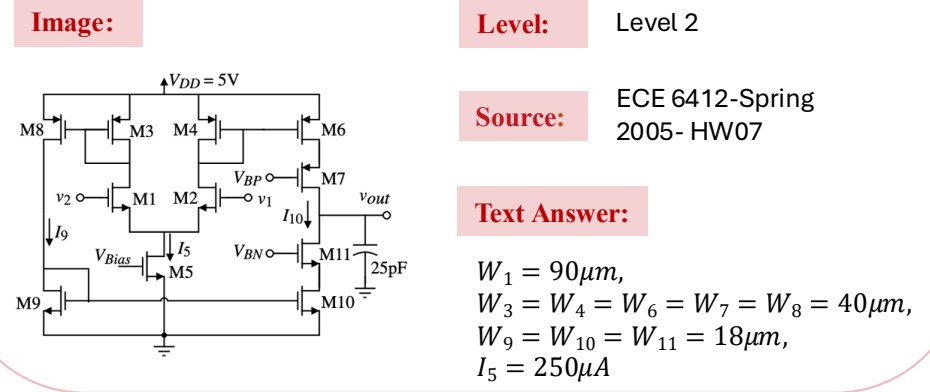

**Level:** Level 2

**Source:** ECE 6412-Spring
2005- HW07

**Text Answer:**

$W_1 = 90\mu m$,
$W_3 = W_4 = W_6 = W_7 = W_8 = 40\mu m$,
$W_9 = W_{10} = W_{11} = 18\mu m$,
$I_5 = 250\mu A$

Example 3: hierarchical design

**Question:**

In the circuit, $R$=2$\Omega$, $L$=1mH, and $C$=0.4$\mu$F. Find the resonant frequency.

**Level:** Level 0.5

**Source:** Fundamentals of Electric Circuits
P674
Example 14.7

**Image:**

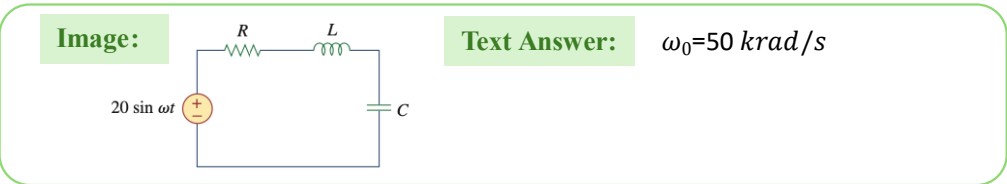

**Text Answer:** $\omega_0$=50 $krad/s$

Example 4: Level 0.5

**Question:**

For the emitter follower output stage shown below, find the value of efficiency when $R_I = \frac{-VEE-VBE}{IQ} = 7.826\text{K}\Omega$ and $V_{CC} = -VEE = 2.5V, VCE = 0.2V, VBE = 0.7V, RL = 10K\Omega$

**Image:**

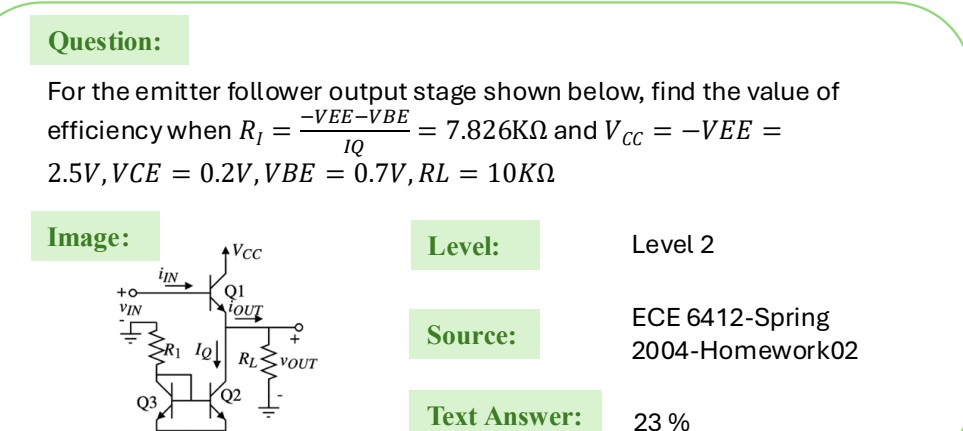

**Level:** Level 2

**Source:** ECE 6412-Spring 2004-Homework02

**Text Answer:** 23 %

Example 5: Level 2

**Question:** Use nodal analysis to determine voltages $V_2$ in the circuit.

**Image:**

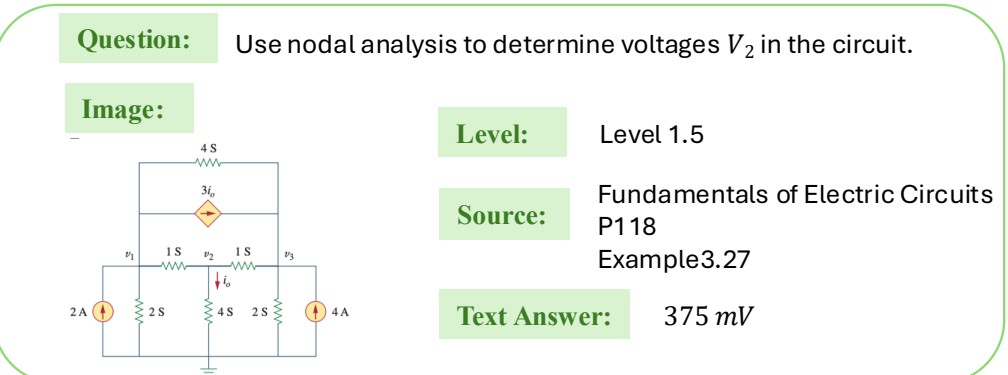

**Level:** Level 1.5

**Source:** Fundamentals of Electric Circuits P118 Example3.27

**Text Answer:** 375 $mV$

Example 6: Level 1.5

**Question:** Determine $v_0$ in the op amp circuit

**Image:**

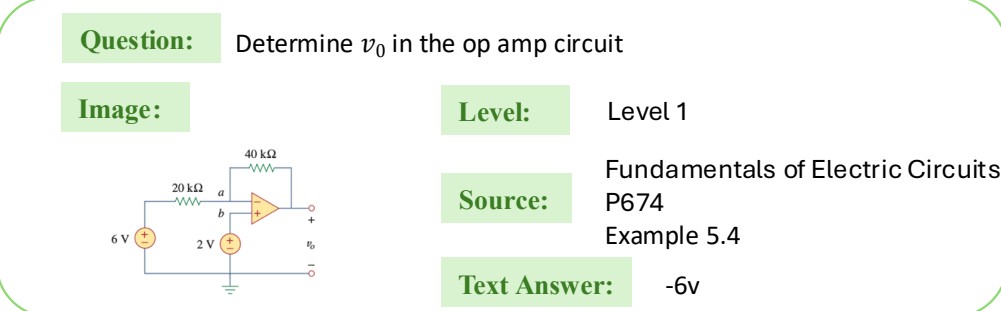

**Level:** Level 1

**Source:** Fundamentals of Electric Circuits P674 Example 5.4

**Text Answer:** -6v

Example 7: Level 1

## A.8 PROMPT TEMPLATES

---

**Prompt Template for Circuit Schematic Synthetic Pipeline**

You are an expert electrical engineer specializing in circuit analysis. Analyze the circuit diagram and solve for the requested symbolic expression.

**Task:** {Main Question}

**Instructions:** 1. Use EXACT component labels as shown in the circuit (e.g., R1, R2, C1, C2, L1, not generic R, C, L) 2. For Laplace domain, use lowercase 's' as the complex frequency variable 3. Use standard impedances: R for resistors, 1/(sC) for capacitors, sL for inductors 4. For op-amps: Apply virtual short (V+ = V-) if in negative feedback, use Ad for gain if specified

**Response Format:** You MUST structure your response exactly as follows:

`<think>`
[Show your reasoning and intermediate steps here] - Identify components and nodes - Intermediate steps - Show equations - Show algebraic manipulation - Any simplification steps
`</think>`
`<answer>`
[Only the final symbolic equation here, e.g., H(s) = ..., Vn1(s) = ..., etc.]
`</answer>`
Make sure to use standard mathematical notation with  for multiplication, / for division, and ^ for powers."

---

**Prompt Template for Block Diagram**

You are an expert in control systems engineering. Your task is to analyze the provided control system block diagram and derive the overall transfer function, C(s)/R(s), in its symbolic form. **Question:** {Main Question} **Instruction:** The final expression must be in terms of 's'. Don't include the high-level symbolic block gains (e.g., G1(s), G2(s), H1(s), F1(s)) in the final answer. Place your final, exact symbolic equation inside `<think>` tags.

`<think>`
$C(s)/R(s) = (s + 1)/(s^2 + 5s + 6)$
`<\think>`
The final answer must be a rational function of 's'.

---

## A.9 LLM USAGE

Large language models were used solely for grammar checking and minor text polishing during manuscript preparation. No LLMs were involved in research ideation, experimental design, data analysis, or substantive writing of the paper.

