# OpenReview forum: "CircuitSense: A Hierarchical MLLM Benchmark Bridging Visual Comprehension and Symbolic Reasoning in Engineering Design Process"
_ICLR.cc/2026/Conference — ICLR 2026 Poster_

### Official Review · Reviewer_Pqrx · 2025-10-14

**Soundness:** 2
**Presentation:** 3
**Contribution:** 1
**Rating:** 4
**Confidence:** 5

**Summary:**

This paper introduces CircuitSense, a benchmark designed to evaluate the circuit-understanding capabilities of multimodal large language models (MLLMs). CircuitSense comprises over 8,000 problems organized into a six-level hierarchical structure. The paper also presents an evaluation of six state-of-the-art MLLMs, revealing that while these models demonstrate strong visual perception, they perform catastrophically poorly in the symbolic reasoning tasks essential for circuit analysis and design.

**Strengths:**

1. The paper is well-written and easy to follow.

2. Addressing multimodal engineering problems is an important and timely topic for current MLLMs.

3. The proposed hierarchical synthetic generation pipeline is interesting.

**Weaknesses:**

1. Lack of discussion on related work (EEE-Bench [1]).
EEE-Bench is a comprehensive multimodal benchmark for electrical and electronics engineering, which is highly relevant to this paper. The authors fail to acknowledge or compare their work against it.

2. Overstated claims and limited novelty.
The paper claims to introduce the first multi-level visual-to-analytical benchmark. However, EEE-Bench has already covered comprehensive circuit analysis problems. The authors do not clearly articulate the differences between CircuitSense and EEE-Bench. In my view, the main tasks (circuit-related problems) in this paper largely overlap with those in EEE-Bench, suggesting limited novelty. Moreover, EEE-Bench addresses a broader domain of electrical and electronics engineering.

3. Insufficient evaluation.
The paper only evaluates six large scale models, which is not enough to provide a comprehensive assessment. More open-source small models and recent proprietary models—such as o3 and o4-mini—should be included.

4. Lack of new insights in the experimental results.
Beyond the introduction of the benchmark itself, the experiments do not provide any particularly novel findings. The observation that MLLMs excel in visual perception but struggle with symbolic reasoning is already well known.

5. Missing implementation details.
The paper does not provide sufficient information about the evaluation setup, such as the prompts used for answer extraction or accuracy calculation or other experimental configurations.

6. Inconsistent complexity-level design.
The benchmark defines multiple complexity levels, but the experimental results do not reflect the expected performance degradation with increasing difficulty. If model performance does not decrease with complexity, it raises questions about the validity and meaning of the defined complexity levels.

Reference

[1] EEE-Bench: A Comprehensive Multimodal Electrical and Electronics Engineering Benchmark. CVPR 2025.

**Questions:**

See weaknesses.

---

> ### Author Response · Authors · 2025-11-20
> **Authore Response (1/3)**
>
> We appreciate the reviewer's valuable comments. Since most questions raised by the reviewer concern the novelty of our contribution and our positioning relative to recent benchmarks, we would like to make an overall clarification on CircuitSense before stepping into the detailed responses. While existing works like EEE-Bench and AMSBench have done a good job establishing baselines for general engineering knowledge, the problem of benchmarking engineering design remains unsolved. For instance, GPT-4o achieves only 46.78% overall accuracy on EEE-Bench, and AMSBench results show that even top models struggle significantly with complex circuit synthesis (Pass@5 < 55%) and fail almost completely at testbench generation.
>
> To address this persistent gap, CircuitSense provides a more **detailed hierarchical evaluation** and **granular error analysis** to pinpoint exactly which stage of reasoning causes these design failures, specifically identifying the critical bottleneck where high perception accuracy (>85%) collapses during *symbolic derivation (<19%)*. Furthermore, recognizing that LLMs are constantly evolving, we introduce a **hierarchical synthetic generation pipeline** that allows for sustainable, contamination-free evaluation and provides infinite high-quality data to improve the performance of weaker models. Finally, we offer specific insights into design-driven analysis, empirically demonstrating that symbolic mathematical reasoning is the fundamental prerequisite for successful circuit synthesis.
>
> Addressing major weaknesses:
>
> ### **W1 & W2: Missing comparison with EEE-Bench and novelty concerns:**
> We thank the reviewer for bringing it to our attention. We have carefully analyzed the EEE-Bench paper and **have updated our related work section to explicitly discuss it in our latest revision**. While we agree there is topical overlap in "circuits," the tasks,  and outputs, the goals of the two benchmarks are fundamentally different. CircuitSense is not a subset of EEE-Bench; it is a vertical deepening of the field that evaluates capabilities EEE-Bench does not touch.
> 1. **Task Divergence:**
> Numerical QA vs. Symbolic Derivation & Design
>     * *EEE-Bench (QA Focus):* EEE-Bench is primarily a Question-Answering benchmark. As noted in their statistics, 72% of questions are Multiple Choice, and the free-form answers are restricted to short text, integers, or floats. For example, their circuit problems ask to calculate a specific voltage value (e.g., "0.75")
>     * *CircuitSense (Design Focus):* CircuitSense evaluates Symbolic Reasoning and Generative Design. We do not ask for a single number; we ask the model to derive the governing symbolic equation (e.g., $H(s) = \frac{1}{1 + sRC}$) or generate a SPICE netlist that meets functional specifications. This is a significantly harder task that mirrors the actual engineering design workflow, which EEE-Bench does not evaluate.
> 2. **Hierarchical Scope vs. Domain Breadth:**
>     * *EEE-Bench (Breadth):* EEE-Bench excels at breadth, covering 10 distinct subdomains from "Electromagnetics" to "Communication Systems". It evaluates the model's general knowledge across the EEE spectrum.
>     * *CircuitSense (Hierarchical Depth):* CircuitSense focuses deeply on the Analog Circuit Design vertical. Uniquely, we organize problems by hierarchy level (Level 0 Resistor to Level 5 System Blocks). This allows us to test if a model can translate a high-level system requirement (e.g., PLL locking range) down to the component level, a cross-level reasoning capability that is absent in EEE-Bench's flat subject organization.
> 3. **Evaluation Methodology: Static Dataset vs. Synthetic Pipeline:**
>     * *EEE-Bench* relies entirely on 2,860 hand-picked/curated problems.
>     * *CircuitSense* introduces a Synthetic Generation Pipeline capable of generating infinite, novel circuit topologies with guaranteed symbolic ground truth. This is a methodological contribution that addresses the "data contamination" issue inherent in curated benchmarks and provides a mechanism for training future models, which a static dataset cannot offer.
>
> While EEE-Bench covers broad EEE topics at a single level, CircuitSense provides deep evaluation across six hierarchical levels. EEE-Bench evaluates breadth of EE knowledge; CircuitSense evaluates depth of mathematical reasoning for circuit design. Both are valuable for different purposes and complementary to each other.

---

> ### Author Response · Authors · 2025-11-20
> **Authore Response (2/3)**
>
> ### **W3: Insufficient evaluation:**
>
> We acknowledge the reviewer's concern about model coverage. We focused on six SOTA models, including the most powerful proprietary and open-source (InternVL3-78B) models available. To make our evaluation even more extensive we expanded our evaluated model and due to reviewers request we added `GPT-4o-mini` (more recent proprietary models) and `Gemma-3-27B` (open-source small model).
>
> New results for cureate multiple choice questions:
>
> | Curated MCQ    | Level 0 (resistor) | Level 1 (RLC) | Level 2 (Small Signal) | Level 3 (Transistor) | Level 4 (Block) |
> | :-------- | :-------: | :-------: | :-------: | :-------: | :-------: |
> | GPT-4o-mini  | 43.88    | 49.58    | 26.03   | 23.20    | 45.83   |
> | Gemma-3-27B | 33.28     | 35.08   | 9.59     | 38.21     | 25.00    |
>
> New results for curtated openended questions:
>
> | Curated Openended    | Level 0 (resistor) | Level 1 (RLC) | Level 2 (Small Signal) | Level 3 (Transistor) | Level 4 (Block) |
> | :-------- | :-------: | :-------: | :-------: | :-------: | :-------: |
> | GPT-4o-mini  | 44.44    | 38.28    | 32.88   | 18.62    | 31.25  |
> | Gemma-3-27B | 24.60     | 25.84   | 8.22     | 10.69     | 8.33   |
>
> New results for synthetic data:
>
> | Synthetic Data    | Level 0 (resistor) | Level 1 (RLC) | Level 2 (Small Signal) | Level 4 (Block) | Level 5 (System) |
> | :-------- | :-------: | :-------: | :-------: | :-------: | :-------: |
> | GPT-4o-mini  | 3.2    | 3.8   | 7.54   | 3.4    | 0.88  |
> | Gemma-3-27B | 0.8     | 3.0   | 6.03     | 4.6     | 4.82   |
>
> We added these two models to our evaluation results as requested and updated the paper in our latest revision with all modifications clearly highlighted in blue for transparency.
>
>
> ### **W4: Lack of new insights:**
> We first thank the reviewer who shares the same opinion as us that MLLMs excel in visual perception but struggle with symbolic reasoning for circuit analysis problems. Previous work like EEE-Bench indeed suggests that “reasoning error and perception error, constituting 50.3% and 26.5% of total error respectively.” This also aligns with our findings. However, addressing this problem requires more detailed error analysis to provide new insight for solving the problem. In EEE-Bench, the majority of error analysis is still qualitative. E.g., “Lack of reasoning. These happen when LLMs can read the image but cannot reason correctly to arrive at the answer. For instance, on the right of Figure 6, the LLM correctly reads the location of contact C but fails to consider the location of resistor R in the overall circuit, thus leading to the wrong answer.” Our paper provides, for the first time, a precise, quantitative diagnosis of why, where, and how this failure manifests in a complex engineering domain. Previous works like EEE-Bench, AMSbench, or MMCircuitEval neither have evaluations on symbolic mathematical understanding nor across the complete hierarchy (from component to system level).
> * **New Insight 1:** The "Memorization Illusion" is Quantified. The 61-point performance drop from textbook curated data to our synthetic data is a novel and critical insight. It proves that high scores on existing benchmarks are largely an "illusion" of memorized textbook patterns, not true reasoning.
> * **New Insight 2:** The Specific Bottleneck is Identified. We don't just say "reasoning is bad." We pinpoint the exact failure point. Our analysis in Table 7 shows that models can identify components (97%) and assign impedances (95%) but fail catastrophically at a specific algebraic step: Output Impedance Derivation (8% accuracy). This is a new, granular insight that pinpoints where future MLLM research may focus.
> * **New Insight 3:** Failure is Not Uniform. We show that models develop "specialized capabilities rather than uniform understanding". For instance, on synthetic data, Gemini-2.5-Pro performs best at Level 2 (38.0%) and Level 5 (35.9%), not at simpler levels. This counter-intuitive finding is a new insight into the "spiky" nature of MLLM capabilities.

---

> ### Author Response · Authors · 2025-11-20
> **Authore Response (3/3)**
>
> ### **W5: Missing implementation details:**
> The information is already included in the appendix:
> * **Prompts:** The exact prompt template used for evaluation is in Appendix A.8. We also added the prompt template for block diagrams in our last revision.
> * **Evaluation Setup:** The detailed evaluation framework, including the use of LLM-as-a-judge (Gemini-2.5-Flash) and our SymPy-based symbolic comparison pipeline (which handles algebraic equivalency), is in Section 3.1.
> * **Model Versions & Parameters:** All model versions, snapshot dates, and inference parameters (temperature, max tokens) are in Appendix A.5.
>
> Please let us know if you are looking for any other details regarding our experiments or evaluation setups. We are glad to provide any details.
>
> ### **W6: Inconsistent complexity-level performance:**
> We respectfully clarify that the "complexity levels" in CircuitSense represent the hierarchical abstraction levels of the engineering design process (from Level 0: Resistor Networks to Level 5: System Block Diagrams), rather than a simple linear "difficulty" scale. The non-monotonic performance observed is not a flaw in benchmark design, but a critical experimental finding that highlights the distinction between memorization and reasoning:
> * **Hierarchy vs. Difficulty:** Real-world analog design requires navigating distinct abstraction layers, not just increasing difficulty. A "System Level" (Level 5) problem involves abstract transfer functions, while a "Transistor Level" (Level 3) problem involves non-linear physics. One is not inherently "harder" than the other; they require different domain knowledge. As noted in the paper, engineers must "systematically decompose architectural blocks into component-level schematics", requiring competency across all levels, not just a progression through them.
>
> * Exposing Memorization: The performance fluctuations reveal where models rely on training data patterns. For example, models perform disproportionately well on curated Level 4 (Block) problems due to their prevalence in textbooks, but this performance collapses on synthetic problems at the same level (Table 6). This "inconsistency" validates that the benchmark successfully disentangles rote retrieval from genuine symbolic derivation.
>
> * Consistent Model Stratification: Despite variations in absolute scores across levels, the relative ranking of models remains statistically robust. Gemini-2.5-Pro consistently outperforms other models across the hierarchy. This consistency confirms that our levels are valid discriminators of model capability, reinforcing our conclusion that true engineering competence requires holistic improvement across the entire design hierarchy rather than optimization for specific sub-tasks.
>
>
> We have made every effort to address Reviewer Pqrx's concerns and hope our response meets your expectations, potentially leading to a higher rating.

---

### Official Review · Reviewer_222z · 2025-10-31

**Soundness:** 3
**Presentation:** 2
**Contribution:** 3
**Rating:** 6
**Confidence:** 3

**Summary:**

This paper presents CircuitSense, a comprehensive hierarchical benchmark for assessing visual-to-symbolic reasoning in circuit understanding.  It focuses on the core engineering ability to translate schematic and system diagrams into mathematical equations, which has not been tested in previous multimodal benchmarks.
CircuitSense has 8,006 problems organized into six hierarchy levels (resistor, RLC, small signal, transistor, block, and system) and three task categories (perception, analysis, and design).  The authors propose a synthetic generation pipeline based on SPICE simulation and Mason's gain formula for block diagrams to ensure ground-truth symbolic equations.
Experiments on six cutting-edge MLLMs (GPT-4o, Gemini-2.5-Pro, Claude-Sonnet-4, InternVL3, Qwen2.5-VL, GLM-4.5V) show that while closed-source models achieve >85% accuracy on perception tasks, their performance on symbolic derivation remains <19%, indicating a severe gap between visual parsing and mathematical reasoning.

**Strengths:**

- Novel Benchmark Scope
CircuitSense combines perception, analysis, and design in a single dataset that reflects real-world circuit design workflows. The six-level hierarchy captures domain progression from physical components to system architecture, and provides a uniquely structured view of how reasoning degrades with abstraction.

-Rigorous Synthetic Generation Pipeline
The dual-stage pipeline (schematic and block diagram) ensures electrical validity and symbolic correctness. The use of SPICE simulation, Lcapy-based symbolic derivation, and Mason's gain formula ensures physics-consistent ground truths, not simply textual labels.

- Comprehensive Evaluation and Diagnostics
Testing six leading MLLMs with curated and synthetic problems (multiple-choice -- open-ended -- symbolic derivation) reveals how pattern-matching fails when symbolic reasoning is needed.

- Fine-grained Task Taxonomy
The benchmark explicitly distinguishes perception subtasks (component detection, connection identification, function classification) from analytical derivation, allowing for the identification of the true bottleneck: symbolic manipulation.

- Strong Domain Validity
Questions are sourced from reputable textbooks (Gray, Razavi, Holberg) and university courses to ensure educational and professional relevance.

**Weaknesses:**

- Limited Novelty in Technical Methodology
Although the domain is novel, the benchmark creation adheres to standard dataset synthesis paradigms (grid-based placement, random topology, template verification).  The originality is primarily in hierarchical symbolic labeling, not algorithmic innovation. Limited Novelty in

-Imbalanced Data Composition
The dataset heavily favors analysis (approximately 7,000 samples) over design (~150 samples), making it difficult to statistically interpret design-task outcomes.

- Synthetic Domain Gap
The significant drop in synthetic performance could be due to distribution shift rather than pure reasoning failure. The paper could better quantify the visual and algebraic difficulty of synthetic problems to address potential domain difference of the synthetic data.

-Limited Evaluation Diversity
All tested models are general MLLMs, with no domain-specific baselines (such as SPICE-aware symbolic solvers or retrieval-augmented MLLMs) included.  This absense makes it unclear whether the failures are the result of poor general reasoning or a lack of domain knowledge.

**Questions:**

- Annotation Quality and Validation
How were the 2,986 curated problems verified?  Were multiple domain experts involved, and how did the annotators agree? Were symbolic derivations manually re-checked after SPICE/Lcapy generation to avoid mismatch errors?

- Synthetic Circuit Generator
How does the generator handle non-linear or transistor-level behavior that symbolic solvers cannot detect? Could the authors explain how "adaptive timeouts" were used to balance completeness and feasibility in symbolic derivation?

- Data Balance and Hierarchical Sampling
The design subset (157 samples) is relatively small.  Do the authors intend to scale this portion using synthetic design tasks (for example, automated topology synthesis with constraints)? Are the counts roughly equal across hierarchy levels, or are some levels (such as transistor or system) underrepresented?

- Table 7 shows a significant drop in output impedance derivation (8%).  Could you provide qualitative examples that demonstrate the specific algebraic or topological misunderstanding? Do models consistently misinterpret node labeling conventions (e.g., sign or orientation errors) or fail algebraic simplification?

- Robustness in Equation Comparisons
The symbolic comparison pipeline (SymPy plus numerical validation) is great. What is the failure rate? Were there instances where algebraically correct but simplified forms (e.g., factorized expressions) were mistakenly classified as incorrect?

- Potential Extensions: Do the authors intend to include temporal behaviors, such as time-domain simulations or frequency-sweep interpretation? Could integration with RAG/agent-based method serve as future baselines?

Appreciate the author's work and contributions to the community!

---

> ### Author Response · Authors · 2025-11-20
> **Author Response (1/5)**
>
> We deeply appreciate the reviewer’s thorough analysis and constructive questions. We appreciate the recognition of our "**novel benchmark scope**" and "**rigorous synthetic generation pipeline**".
>
> Addressing Weaknesses:
>
> ### **W1: Limited novelty in technical methodology:**
> Our technical innovation extends far beyond standard synthesis paradigms. Our pipeline uniquely generates hierarchically-coupled problems that mirror real-world circuit design challenges. The key innovation is the hierarchical design-centric evaluation, and our focus is on real-world circuit engineering design, not just academic physics analysis:
> 1. **Open-ended formulation reflecting real design scenarios:** Engineers don't get multiple-choice options when designing amplifiers or PLLs. They must derive equations from scratch. While existing benchmarks test "Which gain is correct: (a) 10 (b) 20?", we require "Derive H(s) = ..." mirroring actual design practice.
>
> 2. **Hierarchical reasoning requirement:** Real circuit design demands simultaneous consideration across abstraction levels. Designing an op-amp requires understanding:
>     * Component level: transistor small-signal models
>     * Block level: current mirrors, differential pairs
>     * System level: feedback stability, frequency compensation
> Our synthetic pipeline uniquely generates problems requiring this multi-level reasoning.
>
> 3. **Beyond textbook simplification:** textbooks present isolated single-level problems which are numerical and mostly have multiple choices, so the model and guess. Our synthetic generation creates hierarchically-coupled problems which are focused on symbolic derivations and are open-ended. This pipeline is aligned with the design workflow and better evaluates the true knowledge of models.
>
> ### **W2: Imbalanced data composition:**
> We appreciate the reviewer’s observation about the quantitative imbalance between the Analysis ($N=7,043$) and Design ($N=157$) subsets.
> * Our observation is  that the Design tasks possess **sufficient statistical power** to differentiate model capabilities due to their high information density and rigorous evaluation methodology. Unlike the Analysis tasks which often allow for short-form verification, the Design tasks are evaluated via Ngspice simulations using the Skywater 130nm PDK
> * This creates an effectively infinite search space where "random guessing" yields near-zero success; consequently, any positive performance represents a high-confidence signal of genuine competency rather than statistical noise. Furthermore, the effect sizes observed are substantial—for instance, the performance delta between Gemini-2.5-Pro (36.38%) and InternVL3-72B (7.01%) on schematic-level design is distinct enough to establish a clear ranking order without requiring thousands of samples
> * Finally, the Design results are not isolated; they strongly correlate with the Analysis findings, confirming our hypothesis that symbolic reasoning capabilities (validated on the large-scale Analysis set) are a prerequisite for design synthesis.
> Thus, the Design subset serves as a high-fidelity "capstone" evaluation that mirrors real-world engineering, where rigorous design synthesis is naturally less frequent but more computationally demanding than routine analysis.

---

> ### Author Response · Authors · 2025-11-20
> **Authore Response (2/5)**
>
> ### **W3: Synthetic domain gap:**
> We respectfully argue that the performance drop stems from the removal of memorization shortcuts rather than a distributional shift in circuit topology or algebraic complexity. We quantify this through three layers of evidence that validate the synthetic domain's consistency with standard engineering problems:
> * **Topological and Electrical Validity (Section 2.2):** To ensure synthetic circuits do not suffer from "distribution shift" (i.e., becoming physically unrealistic or "weird"), our pipeline enforces strict electrical constraints: eliminating floating nodes, ensuring degree-2 connectivity, and verifying DC operating points via SPICE simulation. Furthermore, we implement adaptive timeouts to filter out circuits with excessive algebraic complexity, ensuring the synthetic dataset remains within the same "solvable" complexity range as the curated textbook problems.
> * **Empirical Validation of Task Alignment (Table 7):** If the synthetic problems represented a fundamental domain shift, models would fail to map the visual topology to the correct analysis domain. However, our granular analysis of 100 synthetic derivation tasks reveals that Gemini-2.5-Pro achieves 95% accuracy on "Impedance Assignment" (converting components to s-domain). This high success rate in the setup phase proves the model recognizes the synthetic tasks as standard circuit analysis problems.
> * **Isolating the Algebraic Reasoning Gap:** The failure occurs strictly during the symbolic manipulation stage. Despite correctly setting up the problem (95%), performance collapses to 8% during "Output Impedance Derivation" and 39% during "Impedance Ratio Formation". This stark contrast within the same problem set quantifies that the bottleneck is not a domain shift (visual or conceptual), but a specific inability to perform multi-step symbolic algebraic reasoning when the "scaffolding" of memorized textbook solutions is removed.
>
> ### **W4: Limited evaluation diversity:**
> * **Domain-specific baseline such as SPICE-aware symbolic solver:** We respectfully clarify that the exclusion of symbolic solvers (e.g., SPICE or Lcapy) as baselines is intentional, as CircuitSense specifically evaluates the visual-to-mathematical reasoning pipeline, which text-based solvers cannot perform without ground-truth netlists. Providing a netlist to a solver would bypass the 'Perception' phase entirely, which is a critical component of the engineering workflow we assess.
>
> * **Reasoning vs. Knowledge:** our results already empirically isolate 'reasoning failures' from 'knowledge gaps' without the need for RAG baselines. Specifically, Closed-source models achieve high accuracy (>85%) on Perception tasks (Component Detection and Topology Identification), proving they possess the requisite domain knowledge to recognize circuit elements. However, their performance collapses to <19% on Analysis tasks involving symbolic derivation. This sharp discrepancy confirms that the failure mode is not a lack of domain knowledge or retrieval capability, but a fundamental deficit in bridging visual comprehension with symbolic mathematical reasoning.
>
> * **Regarding RAG:** A significant portion of our benchmark consists of 5,020 synthetically generated circuits with novel topologies and randomized component values. Because these specific circuit configurations do not exist in any external knowledge base, retrieval mechanisms cannot aid in the core task of ab initio equation derivation, which requires reasoning from first principles rather than memorization.

---

> ### Author Response · Authors · 2025-11-20
> **Authore Response (3/5)**
>
> Answering Detailed Questions:
>
> ### **Q1: Annotation quality and validation:**
> * **Curated dataset:** Two domain experts in circuits and electronics collected problems from authoritative textbooks and datasheets (Gray, Razavi, etc.) where answers were already verified by the academic community. All solutions were cross-referenced with textbook answer keys.
> * **Synthetic dataset:** Our pipeline implements multiple validation layers:
>     * Circuit validity: every generated circuit undergoes SPICE simulation (Ngspice) to verify that there are no floating nodes or invalid topology.
>     * Symbolic verification: ground-truth equations are derived using Lcapy, a well-established library built on SymPy for symbolic circuit analysis. Lcapy is widely used in academia for circuit theory and provides reliable symbolic derivation through Modified Nodal Analysis.
>     * We have verified Lcapy outputs against SPICE AC analysis at 50 frequency points for 100 examples to make sure the pipeline is reliable.
>
> This multi-layer validation ensures mathematical correctness and our testing found no such systematic issues.
>
> ### **Q2: Non-linear/transistor handling and adaptive timeout:**
> * Our synthetic pipeline focuses exclusively on linear circuits: RLC networks, ideal op-amps, and small-signal models with dependent sources. This enables guaranteed symbolic ground-truth derivation. In practical circuit analysis, nonlinear circuits such as transistor-based designs are linearized using small signal models, which can be solved using our pipeline with Lcpay.
>
> * Adaptive Timeout: we compute circuit complexity based on component count, reactive elements, and estimated nodes. Simple circuits receive longer timeouts and complex ones get shorter budgets to prevent stalling. We run each symbolic solve in a subprocess with hard wall-clock timeout, preventing hangs. Our base timeout is 30s for simple circuits (<5 components) and it will decrease as complexity increases. The success rate for circuits <12 components is 92%. This approach balances completeness with computational feasibility. By focusing on solvable linear circuits, we generate thousands of problems with guaranteed correct ground-truth equations while avoiding the computational intractability of complex symbolic algebra.

---

> ### Author Response · Authors · 2025-11-20
> **Author Response (4/5)**
>
> ### **Q3: Data balance and scaling:**
> We thank the reviewer for this insightful question regarding the scale of the design subset and the hierarchical distribution of our data.
> * **Data Balance (Design subset):** While the design subset contains 157 samples, we argue that these tasks possess significantly higher information density and discriminative power than standard analysis questions. We agree that scaling the design subset via synthetic generation is a vital direction for future work. Our current analysis pipeline uses Lcapy for symbolic derivation (linear algebra) to extract ground truth from a known topology. A synthetic design pipeline requires an optimization loop (e.g., simulation-in-the-loop or gradient-based methods) to find the "ground truth" component values that satisfy a randomly generated set of specifications. We plan to extend our synthetic generation pipeline in this direction.
>
> * **Counts and Hierarchy Scaling:** We thank the reviewer for this important observation about the distribution across hierarchy levels. We acknowledge that the sample counts are imbalanced, and we provide the following clarification on why this distribution is both intentional and appropriate for our benchmark's objectives.
> Current Distribution (Analysis Task): Our benchmark contains 1,777 samples at Level 0 (Resistor, 22%), 3,147 at Level 1 (RLC, 45%), 537 at Level 2 (Small Signal, 8%), 795 at Level 3 (Transistor, 11%), 559 at Level 4 (Block, 8%), and 228 at Level 5 (System, 3%).
>     * *Justification for This Distribution:* Low-Level Circuits Require More Samples for Equivalent Statistical Power. The higher concentration at Levels 0-1 reflects the fundamental difference in topological variation across hierarchy levels. Low-level circuits exhibit enormous combinatorial diversity—resistor and RLC networks can be configured in countless topological arrangements with varying component placements, node connections, and circuit structures. In contrast, system-level block diagrams (Level 5) follow more constrained architectural patterns built from standard primitives: feedback loops, cascaded stages, and summing junctions arranged according to established control theory principles. This means that 228 system-level samples can provide equivalent statistical coverage to what requires 3,147 samples at the RLC level, as each high-level sample represents a more constrained design space.
>
>     * *Topological Complexity Decreases with Abstraction Level:* As we move up the hierarchy from components to systems, the design space becomes increasingly structured. Level 0-1 circuits must cover diverse mesh and nodal topologies, series-parallel combinations, bridge configurations, and numerous voltage/current divider arrangements. Level 5 block diagrams, by contrast, are primarily characterized by feedback structure (number of loops, nesting depth) and forward path configuration—a much more limited parameter space. Our sample distribution reflects this reality: we allocate samples proportionally to the topological variation at each level to achieve comparable coverage of the representational space.
>
>     * *Empirical Evidence Confirms Adequate Coverage.* The discriminative power of our benchmark validates this distribution strategy. Table 6 demonstrates that each hierarchy level produces consistent, meaningful performance patterns despite varying sample counts. Consider Level 3 (Transistor) with 795 curated samples: Table 5 shows stable performance differentiation with Gemini-2.5-Pro achieving 81.72%, Claude-Sonnet-4 reaching 72.01%, GPT-4o at 48.80%, while open-source models consistently perform lower (Qwen2.5-VL at 35.94%, InternVL3 at 13.11%). More strikingly, Level 5 (System) contains only 228 samples yet produces **the clearest performance stratification in our entire benchmark**, Gemini-2.5-Pro achieves 35.96% while most other models fall below 10.09%, with this 3.5× performance gap remaining stable across the sample set. If 228 samples were insufficient, we would expect noisy, unstable rankings where small sampling variations cause model performance to fluctuate unpredictably. Instead, we observe consistent, reproducible patterns: the same models that excel at applying Mason's gain formula continue to do so across different system topologies, while models that struggle with symbolic manipulation of rational functions consistently fail. This stability confirms that even our smallest level provides adequate statistical power to reveal genuine differences in mathematical reasoning capabilities rather than measurement noise.

---

> ### Author Response · Authors · 2025-11-20
> **Authore Response (5/5)**
>
> ### **Q4: Output impedance failure analysis:**
> The 8% accuracy on output impedance derivation reveals systematic conceptual misunderstandings, not just algebraic errors. Here's a representative example:
>
> Netlist Connections:
> ```
> R2 1 2 R2
> V1 2 0 step
> C1 1 2 C1
> R1 0 4 R1
> R3 1 4 R3
> ```
> `R2` is designated as the output load. For this circuit, we treat R2 as the output load. In our problem statement, the transfer function is defined as $H(s) = Z_{output}/Z_{total}$, and the model is asked to find $Z_{output}$.
>
> Ground Truth: $R2 || (1/sC1)$
>
> Model's Answer: $R2$
>
> **Correct analysis:**
>
> $Z_{output}$ = impedance between nodes 1 and 2 = $R2 || (1/sC1)$
>
>
> **Model’s systematic errors:**
>
> * Topological misunderstanding: Models treat $Z_{output}$ as just `R2` in isolation, failing to recognize it's the parallel combination with `C1`. They conceptually misunderstand what "*output impedance*" means in the circuit context.
>
> Thus, we believe that most of the output impedance errors are coming from topology misunderstanding instead of misinterpretation of node labeling conventions (e.g., sign or orientation errors) or failing algebraic simplification.
>
> ### **Q5: Equation comparison robustness:**
> SymPy successfully handles factorization, partial fractions, and reciprocal forms, and it also compares two symbolic equations with different methods, such as direct comparison or simplification. It also calculates the difference between two symbolic equations. Even if Sympy can not get the final decision due to the complexity of the equation, the numerical validation will make sure that the pipeline is **robust**.
>
> ### **Q6: Future extensions:**
>
> Absolutely! We are planning to extend the pipeline so that it covers time-domain analysis and frequency-domain analysis more broadly. The final goal is to have an agent that can use these symbolic equations, textbook/datasheet retrieval through RAG, and simulation in-the-loop to perform circuit design with iterative refinement faster and more accurately.

---

### Official Review · Reviewer_bBsZ · 2025-10-31

**Soundness:** 2
**Presentation:** 3
**Contribution:** 3
**Rating:** 6
**Confidence:** 3

**Summary:**

This paper introduces CircuitSense, a hierarchical benchmark designed to evaluate multimodal large language models on circuit understanding and symbolic reasoning tasks across over 8,000 problems. The study shows that while models perform well on visual perception tasks, they struggle significantly with symbolic derivation, highlighting a critical gap between visual comprehension and mathematical reasoning in engineering design.

**Strengths:**

- The dataset is highly diverse, encompassing various difficulty levels and a wide range of analog question types. This diversity could significantly benefit future research on large language models (LLMs) in analog reasoning.

- The experimental evaluation is thorough, involving multiple LLMs with different capability levels, providing a comprehensive comparison.

**Weaknesses:**

- Although the work is valuable, several existing benchmarks—such as AMSBench and MMCircuitEval—appear to share similar objectives and formulations. The differences between this work and those benchmarks are not clearly articulated.

- In the supplementary materials, there are multiple zero-byte files (e.g., in the Perception/func directory), which may indicate missing or corrupted data.

Typos:

In Table 4, “GPT-4O” should be corrected to “GPT-4o”.

**Questions:**

How is the correctness of the LLM’s answers evaluated? For example, in the Synthetic Example Q5, there seem to be multiple mathematically correct equations that can represent the same diagram. How is such ambiguity handled during evaluation?

---

> ### Author Response · Authors · 2025-11-20
> **Authore Response**
>
> We thank the reviewer for their careful evaluation and for identifying important issues that we will address.
>
> Addressing Weaknesses:
>
> ### **W1: Differentiation from existing benchmarks (AMSBench, MMCircuitEval):**
> We appreciate this critical point and will clarify the key distinctions:
> * **AMSBench (Shi et al., 2025):**
>     * The main focus is on perception tasks. 6k out of 8k samples are focused on perception tasks. However, CircuitSense focuses primarily on the Analysis task.
>     * Does not have hierarchical organization, such as block diagram system level questions with symbolic transfer function ground truth.
>     * Focuses only on multiple-choice questions, which are not aligned with typical real-world design problems where there are no other answers for selection, and is typically in an open-ended format. The accuracy on most of the benchmarks (perception and analysis) is already very high (~%90). Their high accuracy results indicate that vision/perception is not the problem of solving analog circuit design problems. In contrast, our benchmark result indicates that symbolic analysis (such as hierarchical reasoning) is a key element that current models lack. This has not been discussed or covered by previous benchmarks.
> * **MMCircuitEval:**
>     * Also does not evaluate the symbolic equations derivation
>     * Does not have a hierarchical organization (from component to system), as this is an important characteristic in the design workflow. Real circuit design demands simultaneous consideration across abstraction levels. Our synthetic pipeline uniquely generates problems requiring this multi-level reasoning, which previous works completely lack.
>
> AMSbench and MMCircuitEval gathered data from textbooks and datasheets, which can suffer from data contamination and may lead to high accuracy. We proposed a synthetic pipeline that generated new circuit-question-answer pairs that models have not seen before. Our benchmark is more sustainable compared to existing benchmarks that heavily rely on existing online data.
>
> ### **W2: Zero-byte files in supplementary materials:**
> We sincerely apologize for this technical issue. We have identified and fixed the corrupted files. Due to the file size limit for supplementary material (only 100MB), we uploaded the data into an anonymized HuggingFace account: [Link](https://huggingface.co/datasets/anony12345567/circuit_sense_iclr/tree/main)
>
> ### **Typos:**
> We thank the reviewer for pointing that out. We have fixed the typo in our latest revision.
>
> ---
>
>
> Answering the Question:
>
> ### **Q1: Evaluation of correctness for multiple valid representations:**
> We handle this comprehensively:
> * Symbolic equivalence checking via SymPy: We parse both ground-truth and predicted equations into symbolic expression trees and verify mathematical equivalence through 1. Algebraic simplification to canonical form and 2. Symbolic subtraction (should equal zero for equivalent expressions)
> * Numerical validation: For complex expressions where symbolic verification is computationally intractable, we evaluate both expressions at 100 random complex frequency points and check numerical agreement
>
> So for synthetic example Q5 any algebraically equivalent form (e.g., factored, expanded, or rearranged) would be marked correct (handled with SymPy library).

---

### Official Review · Reviewer_h46k · 2025-11-01

**Soundness:** 3
**Presentation:** 3
**Contribution:** 4
**Rating:** 8
**Confidence:** 4

**Summary:**

The paper introduces CircuitSense, a large-scale (8,006 problems across six hierarchical abstraction levels spread across Perception, Analysis, and Design problems) benchmark designed to evaluate visual mathematical reasoning in circuit analysis and design. This remains an under-explored domain for multimodal large language models (MLLMs).

The authors construct a hierarchical synthetic data generation pipeline capable of automatically producing novel circuit schematics and block diagrams with guaranteed symbolic ground-truth equations. They achieved this via SPICE simulation and symbolic analysis via SymPy and Lcapy. This allows systematic testing of models’ ability to convert visual circuit representations into symbolic transfer functions and equations, which is a step beyond conventional recognition or numerical tasks that exist in the current VLM’s.

Evals across six leading MLLMs such as GPT-4o, Gemini-2.5-Pro, Claude-Sonnet-4, InternVL3, etc. reveal a sharp contrast between perception and reasoning: while models achieve >85% accuracy on component recognition and topology understanding, symbolic derivation accuracy collapses below 19%, with catastrophic failure on novel synthetic circuits. This demonstrates the baseline reliability of the benchmark in evaluating SOTA models on these tasks.

**Strengths:**

- The paper tackles an unexplored domain: hierarchical visual symbolic reasoning in circuit systems, with high-quality richly annotated data. It seems to be the first benchmark to evaluate how well AI systems connect circuit visuals to mathematical models. Prior work such as ChipVQA only partially evaluates this capability.
- Six abstraction levels mirror real analog design processes and provide a structured way to locate where model failures occur
- SDG pipeline and datasets are open-sourced and readily reproducible and could constitute a major contribution to the community in solving the data scarcity problem
- Failure analysis is fine-grained, highlighting, for example, precise points of reasoning breakdown (output impedance and input impedance)

**Weaknesses:**

- Sim2Real gap remains: while the synthetic generation pipeline is well engineered, the circuits are constrained to 12-15 components; data sourcing from internet data might result in contamination
- The benchmark relies on Gemini as the judge, which may result in potential biases toward the Gemini family
- The paper could benefit in an analysis on cross-domain transfer; for example, training on schematic-level circuits and testing on block diagrams
- Lastly, the paper would benefit in adding human baseline performance

**Questions:**

- Beyond algebraic identity via SymPy, have you considered functionally equivalent but structurally different forms? This might influence the reported accuracy at <19%.

- How closely do generated circuits reflect real textbook or industrial designs?

- How did you ensure fairness when benchmarking Gemini models using Gemini as a judge?

- Could CircuitSense outputs be integrated into simulation-based tools to test end-to-end utility for analog designers?

---

> ### Author Response · Authors · 2025-11-20
> **Author Response**
>
> We thank the reviewer for their thorough evaluation and constructive feedback.
>
> Addressing Weaknesses:
>
> ### **W1: Sim2Real gap, component constraints, and data contamination:**
> * We acknowledge the 12-15 component limitation stems from computational constraints in symbolic equation derivation. However, this range perfectly covers most fundamental analog building blocks found in textbooks. It is worth mentioning that CircuitSense has different hierarchy levels, and this component limitation only exists at the lowest level. For system-level, it can go up to **25 blocks**, which is **well aligned with real-world system design tasks**. Although real-world analog applications may include circuits with more components, they often can be broken into sub-circuit blocks. Performing circuit analysis on a design with <15 components can already have many practical uses. Recent work like LaMAGIC [1] demonstrates that even 6-component circuits pose significant challenges for automated design. Our 12-15 component range represents a substantial advance in complexity for symbolic analysis benchmarks.
> * Regarding data contamination, we explicitly separated our evaluation into curated (potentially contaminated) and synthetic (**guaranteed novel**) problems. The dramatic performance drop on synthetic circuits (Table 6: 80.71% → 19.06% for Gemini-2.5-Pro) demonstrates that contamination cannot explain the poor mathematical reasoning performance.
>
>
>
> ### **W2: Gemini as judge bias:**
> To address this, we implemented multiple safeguards.
> * For symbolic equations, we use SymPy for mathematical equivalence checking, not LLM judgment. For multiple-choice questions, we use exact matching.
> * LLM-as-judge is only used for open-ended short answers (only a subset of CircuitSense) where it is mostly useful for equivariant representation and unit conversion.
>
> ### **W3: Cross-domain transfer analysis:**
> This is an excellent suggestion. While our current evaluation shows performance across levels (Table 5-6), we didn't explicitly test transfer learning. We will consider this in future works (testing models fine-tuned on Level 0-2 circuits on Level 3-5 problems). In this paper, we focused on proposing a benchmark to evaluate the current model’s visual reasoning ability in circuits.
>
> ### **W4: Human baseline performance:**
> We agree this would strengthen the benchmark. As future work, we plan to collect human performance data from electrical engineering students at different levels (undergraduate/graduate) on a subset of 500 problems across all categories.
>
> ---
>
> Addressing Questions:
> ### **Q1: Functionally equivalent forms:**
> Excellent point. We handle this through our two-tier verification:
> * Symbolic verification (SymPy): algebraic identity
> * Numerical validation: evaluates both expressions at 100 random complex frequency points. In this way, we address functionally equivalent but structurally different forms.
>
> ### **Q2: Real-world design similarity:**
> Our synthetic pipeline randomly connects different components and puts them in a `n` by `m` grid, so it resembles the textbook circuit figures. The questions for synthetic circuits are also similar to real-world problems. E.g., “What is the transfer function from one node to another?”. Furthermore, some of our design questions have circuits drawn in Cadence by circuit design engineers, which is an industry-standard schematic editing tool.
>
> ### **Q3: Fairness in Gemini benchmarking:**
> As noted in W2, our primary evaluation uses deterministic methods (SymPy, exact matching). For the subset requiring LLM judgment, we use structured prompts focusing on mathematical equivalence. We also used different Gemini models. For judging, we use Gemini-Flash, but to evaluate Gemini-Pro, which may be less biased on each other.
>
> ### **Q4: Integration with simulation tools:**
> We generate standard netlists that are compatible with standard simulation tools such as Ngspice, Cadence Spectre, and HSPICE. And finally, for design questions in our benchmark, we already treat LLM as a sizing agent that connects with Ngspice end-to-end by letting LLM generate a SPICE netlist with sizing and bias voltage.
>
> [1] Chang et. al., LaMAGIC: Language-Model-based Topology Generation for Analog Integrated Circuits

---

### Author Response · Authors · 2025-11-21
**General Author Response**

We sincerely thank all reviewers for their time, effort, and constructive feedback on CircuitSense. We are grateful for the reviewers’ recognition of our work’s contribution and impact potential. The reviewers identified major contributions of our works:
1. **First benchmark requiring symbolic equation derivation from visual circuits**
    * Quote from Reviewer “h46k”: "It seems to be the first benchmark to evaluate how well AI systems connect circuit visuals to mathematical models."
    * Quote from Reviewer "222z": "focuses on the core engineering ability to translate schematic and system diagrams into mathematical equations, which has not been tested in previous multimodal benchmarks."
    * Quote from Reviewer "222z": "CircuitSense combines perception, analysis, and design in a single dataset that reflects real-world circuit design workflows."
2. **Rigorous hierarchical synthetic generation pipeline with guaranteed ground truth**
    * Quote from Reviewer “h46k”: "SDG pipeline and datasets are open-sourced and readily reproducible and could constitute a major contribution to the community in solving the data scarcity problem."
    * Quote from Reviewer "222z": "The dual-stage pipeline (schematic and block diagram) ensures electrical validity and symbolic correctness. The use of SPICE simulation, Lcapy-based symbolic derivation, and Mason's gain formula ensures physics-consistent ground truths."
    * Quote from Reviewer “Pqrx”: "The proposed hierarchical synthetic generation pipeline is interesting."
    * Quote from Reviewer “222z”: "Rigorous Synthetic Generation Pipeline... ensures physics-consistent ground truths, not simply textual labels."
3. **Comprehensive hierarchical evaluation revealing critical reasoning gaps**
    * Quote from Reviewer “h46k”: "Six abstraction levels mirror real analog design processes and provide a structured way to locate where model failures occur."
    * Quote from Reviewer “h46k”: "Failure analysis is fine-grained, highlighting, for example, precise points of reasoning breakdown (output impedance and input impedance)."
    * Quote from Reviewer “222z”: "The six-level hierarchy captures domain progression from physical components to system architecture, and provides a uniquely structured view of how reasoning degrades with abstraction."
    * Quote from Reviewer “bBsZ”: "The dataset is highly diverse, encompassing various difficulty levels and a wide range of analog question types. This diversity could significantly benefit future research."
    * Quote from Reviewer “Pqrx”: “Addressing multimodal engineering problems is an important and timely topic for current MLLMs.

We have thoroughly addressed all reviewer comments and hope our responses have resolved the raised concerns. We welcome any follow-up questions and have revised the manuscript with all changes clearly marked in blue.

---

### Meta-Review · Area_Chair_Snyi · 2026-01-05

**Summary:**

This paper introduces CircuitSense, a hierarchical benchmark for circuit understanding that evaluates the full engineering workflow, including perception, analysis, and design, with a particular emphasis on deriving symbolic equations from visual circuit schematics and block diagrams. The benchmark includes >8,006 problems spanning six abstraction levels. A key empirical finding is that current MLLMs perform strongly on visual perception tasks but perform poorly on symbolic derivation, with the gap widening on novel synthetic circuits.

After the rebuttal, the main concern of reviewers centers on novelty/positioning relative to existing benchmarks and evaluation details. Given that reviewers were broadly positive about the benchmark's utility, that the authors' rebuttal clarifies positioning and strengthens the evaluation description, and that CircuitSense looks like a valuable benchmark contribution with a clear diagnostic angle, I recommend acceptance.

**Reviewer Concerns:**

Initial reviewer concerns focused on:
- Novelty and positioning relative to existing benchmarks.
- Evaluation details
- Model coverage, with requests to include additional baselines/models.
- Design evaluation size, since the design subset is relatively small.
- Reproducibility issues.
- Sim-to-real/generalization concerns, given constraints in the synthetic circuit generation.
- Requests for more practical baselines and human baselines.

During rebuttal, the authors addressed these points through clarifications and (where feasible) additional experiments. Overall, the response resolves or mitigates the major concerns to a reasonable extent.

**Reviewer Scores:**

The reviewer who give the least score (4) did not involve in the discussion, so it is difficult to predict if the score would change. The other reviewers are positive to the paper and keep their high scores.

---

### Decision · Program_Chairs · 2026-01-26

Accept (Poster)